# PIDDosome-induced p53-dependent ploidy restriction facilitates hepatocarcinogenesis

Valentina C Sladky[1] (iD), Katja Knapp[1], Tamas G Szabo[1], Vincent Z Braun[1], Laura Bongiovanni[2], Hilda van den Bos[3], Diana CJ Spierings[3], Bart Westendorp[2] (iD), Ana Curinha[4], Tatjana Stojakovic[5], Hubert Scharnagl[6], Gerald Timelthaler[7], Kaoru Tsuchia[8], Matthias Pinter[9], Georg Semmler[9], Floris Foijer[3] (iD), Alain de Bruin[2,10] (iD), Thomas Reiberger[9,11,12], Nataliya Rohr-Udilova[9] & Andreas Villunger[1,11,12,*] (iD)

## Abstract

**Polyploidization frequently precedes tumorigenesis but also occurs during normal development in several tissues. Hepatocyte ploidy is controlled by the PIDDosome during development and regeneration. This multi-protein complex is activated by supernumerary centrosomes to induce p53 and restrict proliferation of polyploid cells, otherwise prone for chromosomal instability. PIDDosome deficiency in the liver results in drastically increased polyploidy. To investigate PIDDosome-induced p53-activation in the pathogenesis of liver cancer, we chemically induced hepatocellular carcinoma (HCC) in mice. Strikingly, PIDDosome deficiency reduced tumor number and burden, despite the inability to activate p53 in polyploid cells. Liver tumors arise primarily from cells with low ploidy, indicating an intrinsic pro-tumorigenic effect of PIDDosome-mediated ploidy restriction. These data suggest that hyperpolyploidization caused by PIDDosome deficiency protects from HCC. Moreover, high tumor cell density, as a surrogate marker of low ploidy, predicts poor survival of HCC patients receiving liver transplantation. Together, we show that the PIDDosome is a potential therapeutic target to manipulate hepatocyte polyploidization for HCC prevention and that tumor cell density may serve as a novel prognostic marker for recurrence-free survival in HCC patients.**

**Keywords** caspase-2; hepatocellular carcinoma; p53; PIDD1; polyploidy
**Subject Categories** Cancer; Cell Cycle; Signal Transduction

## Introduction

Hepatocellular carcinoma (HCC) represents the sixth most common cancer globally and is ranked as fourth leading cause of cancer-related mortality, with steadily increasing incidence (Villanueva, 2019). HCC almost exclusively develops on the background of chronic liver disease such as viral hepatitis or steatohepatitis and cirrhosis (Villanueva, 2019). Due to common late presentation with extensive disease, curative therapy is challenging and can only be provided by liver resection or transplantation (Sapisochin & Bruix, 2017). Thus, a detailed understanding of the molecular mechanisms underlying hepatocarcinogenesis is essential for disease prevention as is the identification of new therapeutic targets.

Liver cells are largely polyploid, a feature usually linked to chromosomal instability and subsequent aneuploidy, which is to some degree seen already in healthy hepatocytes (Knouse *et al*, 2014; Sladky *et al*, 2020). In most other cell types, polyploidy and chromosomal instability are considered to put cells at risk for malignant transformation (Ganem *et al*, 2007; Lens & Medema, 2019). Of note, more than one third of all human cancers is predicted to arise from tetraploid intermediates highlighting the significance of tight ploidy control (Zack *et al*, 2013).

1  Institute of Developmental Immunology, Biocenter, Medical University of Innsbruck, Innsbruck, Austria
2  Department of Biomolecular Health Sciences, Faculty of Veterinary Medicine, Utrecht University, Utrecht, The Netherlands
3  European Research Institute for the Biology of Ageing, University of Groningen, University Medical Center Groningen, Groningen, The Netherlands
4  Institute of Pathophysiology, Biocenter, Medical University of Innsbruck, Innsbruck, Austria
5  Clinical Institute of Medical and Chemical Laboratory Diagnostics, University Hospital Graz, Graz, Austria
6  Clinical Institute of Medical and Chemical Laboratory Diagnostics, Medical University of Graz, Graz, Austria
7  Institute for Cancer Research, Internal Medicine I, Medical University of Vienna, Vienna, Austria
8  Department of Gastroenterology & Hepatology, Musashino Red Cross Hospital, Tokyo, Japan
9  Division of Gastroenterology and Hepatology, Department of Medicine III, Medical University of Vienna, Vienna, Austria
10  Department Pediatrics, University Medical Center Groningen, University Groningen, Groningen, The Netherlands
11  Ludwig Boltzmann Institute for Rare and Undiagnosed Diseases (LBI-RUD), Vienna, Austria
12  CeMM Research Center for Molecular Medicine of the Austrian Academy of Sciences, Vienna, Austria
*Corresponding author. Tel: +43 512 9003 70380; E-mail: andreas.villunger@i-med.ac.at

One of the most frequently inactivated genes in HCC is the major tumor suppressor gene *TP53,* which is mutated in about 30% of all patients (Lee, 2015). p53 is activated by various triggers, including DNA damage, extended mitotic timing or ploidy increases and extra centrosomes, complex structures that organize the mitotic spindle of animal cells. Active p53 restricts proliferation, induces cell death, and orchestrates DNA repair to control genomic integrity and to prevent cancer (Kastenhuber & Lowe, 2017). Remarkably though, p53-induced cell death can also promote liver cancer as loss of functional parenchyma fuels compensatory proliferation in the presence of DNA damage (Qiu *et al*, 2011). Moreover, p53-associated activation of p21-induced cell cycle arrest can allow the survival of cells with altered genomes (Wang *et al*, 1997; De La Cueva *et al*, 2006; Jackson *et al*, 2012). Thus, p53 activation is not unambiguously tumor suppressive and its effects on carcinogenesis are clearly context dependent (Qiu *et al*, 2011).

Among all the different signals activating p53, it is the presence of supernumerary centrosomes that indicate a desired but also an aberrant increase in cellular ploidy (Andreassen *et al*, 2001; Godinho & Pellman, 2014). Extra centrosomes can potentially promote chromosomal instability in subsequent multipolar cell divisions, priming cells for aneuploidy (Bakhoum & Compton, 2009; Godinho & Pellman, 2014; Levine *et al*, 2017). Consistently, aberrant centrosome numbers are frequently found in human cancer (Chan, 2011; Godinho & Pellman, 2014). The molecular mechanism linking extra centrosomes to p53 activation involves the PIDDosome, a multi-protein complex which is formed by PIDD1 and RAIDD to activate caspase-2 (Tinel & Tschopp, 2004; Fava *et al*, 2017), a protease with documented tumor-suppressive capacity (Ho *et al*, 2009; Ribe *et al*, 2012; Parsons *et al*, 2013; Puccini *et al*, 2013a) as well as regulatory roles in liver metabolism (Wilson *et al*, 2015; Kim *et al*, 2018).

Caspases are best-known for their roles in inflammation and cell death (Van Opdenbosch & Lamkanfi, 2019). The PIDDosome and caspase-2, however, are essential to control cellular ploidy by surveillance of centrosome numbers (Fava *et al*, 2017). The presence of supernumerary centrosomes indicates cell cycle defects leading to polyploidization, such as aborted mitoses or incomplete cytokinesis, and is sensed by PIDD1, residing at the mother centriole (Fava *et al*, 2017). Formation of the PIDDosome complex leads to caspase-2 activation, which in turn inactivates the E3-ligase MDM2 that targets p53 for proteasomal degradation. This stabilizes p53 resulting in transcriptional induction of p21 to arrest the cell cycle (Oliver *et al*, 2011; Fava *et al*, 2017). Hence, the PIDDosome prevents proliferation of cells amplifying centrosomes or following a tetraploidization step, both events associated with tumor initiation and progression (Ganem *et al*, 2007; LoMastro & Holland, 2019; Lens & Medema, 2019). As such, the PIDDosome should exert tumor-suppressive functions that may explain observations of increased rates of tumor formation in mice lacking caspase-2, challenged by oncogenic drivers such as MYC or ERBB2, or concomitant ATM loss (Ho *et al*, 2009; Manzl *et al*, 2012; Parsons *et al*, 2013; Puccini *et al*, 2013b).

We recently showed that the PIDDosome controls physiological polyploidization in the liver. Hepatocytes utilize this multiprotein complex to control the upper limit of liver ploidy during development and regeneration. In the liver, expression of the centrosome-sensing component PIDD1 and the executioner

component, caspase-2, is coupled to proliferation, as it is controlled by antagonizing members of the E2F transcription factor family, known for their essential role in ploidy control (Conner *et al*, 2003; Pandit *et al*, 2012; Chen *et al*, 2012). The activating transcription factor E2F1 induces *Pidd1* and *Casp2* expression which allows induction of the PIDDosome-p53-p21 axis to restrict further proliferation of polyploid hepatocytes. This is counteracted by E2F7 and E2F8 repressing PIDDosome transcription to allow for proliferation in a polyploid state in the presence of extra centrosomes (Sladky *et al*, 2020). Loss of either PIDDosome component drastically increases hepatocyte ploidy. While liver function seems unaffected, the degree of aneuploidy is relatively higher as this is linked to the basal ploidy state, but not limited by PIDDosome-induced p53 activity itself (Sladky *et al*, 2020).

Based on the above, we hypothesized that the PIDDosome pathway is not only upstream of the tumor suppressor p53 but also regulates the degree of hepatocyte polyploidy and aneuploidy in response to extra centrosomes to prevent carcinogenesis (Serçin *et al*, 2016; Levine *et al*, 2017). Hence, we investigated experimentally whether PIDDosome-mediated p53 activation limits HCC development, progression, or tumor karyotype evolution. Unexpectedly, our results document a tumor-promoting role for p53-dependent ploidy restriction in liver cancer and that the proliferation of tetraploid cells does not drive aneuploidy in HCC. Furthermore, our findings identify the PIDDosome as a potential drug-target in HCC.

# Results

## PIDDosome-induced p53 activation facilitates DEN-induced liver tumorigenesis

The PIDDosome plays a crucial role in restricting liver ploidy during development and regeneration in hepatocytes acting upstream of p53 in response to cytokinesis failure-induced centrosome accumulation (Fava *et al*, 2017; Sladky *et al*, 2020). Thus, we wanted to test whether PIDDosome-mediated p53 activation limits tumorigenesis in the liver. To address this question, we used the DEN-driven model for chemically induced HCC in wt and mice lacking either *Casp2, Raidd/Cradd,* or *Pidd1*. The livers were isolated and analyzed 10 months after DEN injection (Fig 1A). Liver tumors, as hepatocellular adenoma and carcinoma, developed in all genotypes and in all mice (except for one) and were associated with the presence of pre-neoplastic lesions in the surrounding non-tumorous tissue (Fig EV1A–C). Serum levels of key hepatic parameters such as aminotransferases (ALT, AST), total bilirubin, and urea were comparable between genotypes while lower cholesterol and triglyceride levels were found in PIDDosome-deficient animals (Fig EV1D), a finding in line with a proposed role of caspase-2 in *de novo* lipogenesis (Kim *et al*, 2018). The grade of inflammation and types of immune infiltrates were similar across genotypes, although more plasma cells were found in RAIDD-deficient tumors (Fig EV1E and F).

Despite similar malignancy in terms of infiltrating growth or tumor necrosis across genotypes, wt livers showed a significantly higher number of surface tumors per liver when compared to PIDDosome-deficient mice (Fig 1B and E). Moreover, histopathological examination of non-tumorous liver tissue revealed that the

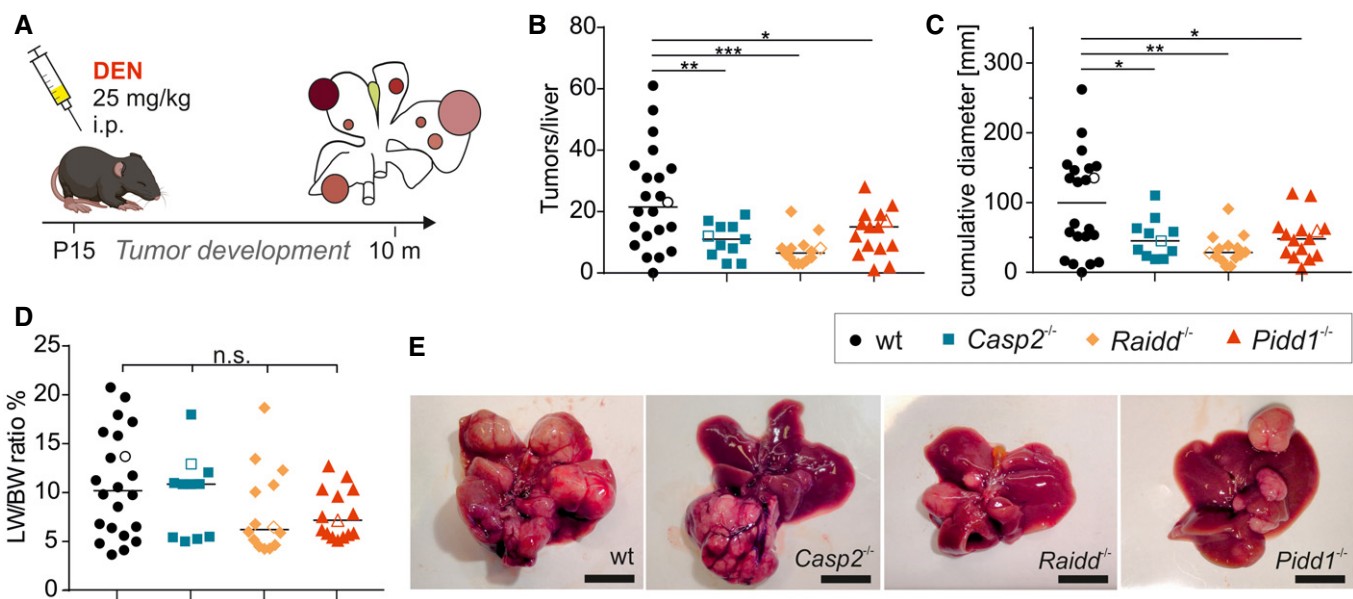

**Figure 1. Loss of PIDDosome function reduces tumor formation in DEN-induced HCC.**

A Male wt and PIDDosome-deficient mice were injected with 25 mg/kg DEN at the age of 15 days, and their livers were analyzed 10 months later.
B Numbers of surface tumors per liver were counted.
C The cumulative diameter of all surface tumors per liver per mouse reflects the overall tumor load.
D Weight of the livers including tumor mass relative to the body weight (LW/BW ratio) was recorded.
E Representative pictures of total livers with tumors of the indicated genotypes.

Data information: Open symbols in graphs (B–D) represent the mice shown in (E). wt $n = 22$, $Casp2^{-/-}$ $n = 11$, $Raidd^{-/-}$ $n = 14$, $Pidd1^{-/-}$ $n = 15$. The line in (B–D) represents the median value of each group, and statistical significance was determined by one-way ANOVA with multiplicity correction (Sidak–Holm) and defined as *$P \leq 0.05$, **$P \leq 0.01$, ***$P \leq 0.001$; scale bar in (E) represents 1 cm.

incidence of pre-neoplastic lesions was also higher in wt mice (Fig EV1C). In line, the total tumor load was clearly higher in wt animals, reflected here by the cumulative diameter of surface tumors (Fig 1C). Interestingly, large tumors (> 8 mm) tended to occur more frequently in PIDDosome-deficient mice (Fig EV1G), which likely explains the comparable liver to body weight ratios (Fig 1D).

### DEN-induced liver tumors upregulate caspase-2 expression

Adult mouse hepatocytes express neither caspase-2 protein nor appreciable levels of *Pidd1* mRNA (Sladky *et al*, 2020), raising the question how the absence of the PIDDosome may affect tumorigenesis. Hence, we tested mRNA and protein levels of caspase-2 and RAIDD as well as *Pidd1* transcripts in DEN-induced HCC. Consistent with our previous results, caspase-2 protein levels were found to be barely detectable in non-tumorous, presumably healthy and largely quiescent liver tissue. Interestingly, however, caspase-2 protein was found readily expressed in all wt tumors tested, as was p53 itself (Fig 2A). The same trends were observed for transcript levels of *Casp2* and *Pidd1* in tumor tissue (Fig EV2A). Due to the lack of suitable antibodies, we could not confirm increased PIDD1 protein expression in tumor tissue. Notably, we found consistent and similar RAIDD expression in both healthy liver and tumor tissue (Figs 2A and EV2A).

As the PIDDosome signals upstream of p53 to limit cell cycle progression in response to extra centrosomes, we next assessed

proliferation rates in liver tumors and non-tumorous liver tissue by Ki67 staining for flow cytometry and immunohistochemistry. In hepatic non-tumorous tissue, proliferation rates were low but similar in all genotypes tested. Despite appreciable differences in tumor size between wt and PIDDosome-deficient mice, Ki67 staining indicated that tumor-proliferation rates were in a comparable range, with a weak increase notable by flow cytometry only in the absence of caspase-2 (Fig 2B). Immunohistochemistry, however, failed to confirm significant increases in Ki67+ tumor cells in $Casp2^{-/-}$ mice (Fig EV2B). Also, mRNA expression of the cancer stem cell (CSC) markers CD90 (Thy1) and CD133 (Prom1) (Yamashita & Wang, 2013) was not substantially different between wt and $Casp2^{-/-}$ liver tumors (Fig EV2C). This suggests that the PIDDosome-regulated p53 response facilitates tumor initiation but not tumor progression. Hence, we closely analyzed PIDDosome-dependent p53 activation at the time of DEN injection. 15-day-old wt and $Pidd1^{-/-}$ mice were injected with DEN or PBS, as a control (Fig 2C). After 8h, both genotypes showed highly comparable induction of the DNA-damage marker phospho-histone γH2AX, as well as of p53 itself. 48h after injection, the levels of γH2AX and p53 were still comparable between genotypes suggesting that the DNA-damage response and p53 activation do not depend on PIDD1 under these conditions (Fig 2D). Of note, at that time, the degree of polyploidization, assessed by the percentage of cells showing binucleation, is still low (Figs 2E and EV2D). Alternatively, DEN-injected mice were weaned at P19 and analyzed 5 days later. At this time point, the PIDDosome is activated by the presence of extra centrosomes that accumulate

during the programmed polyploidization process (Sladky *et al*, 2020). Consistent with prior results, PIDD1-dependent p53 activation was seen only at this time point. Despite a clear background signal for p53 caused by persisting DNA damage, the levels of p53 were now clearly lower in DEN-treated *Pidd1*$^{-/-}$ mice and barely detectably in PBS controls (Fig 2D). Together, this shows that the

PIDDosome does not contribute to p53 activation after DEN injection. DNA damage was reported to induce centrosome amplification in certain conditions (Dodson *et al*, 2004). To test whether DEN could induce PIDDosome activation via this route, we assessed centriole number at the time of injection and 48h later by immunofluorescence staining of the centriolar protein CP110.

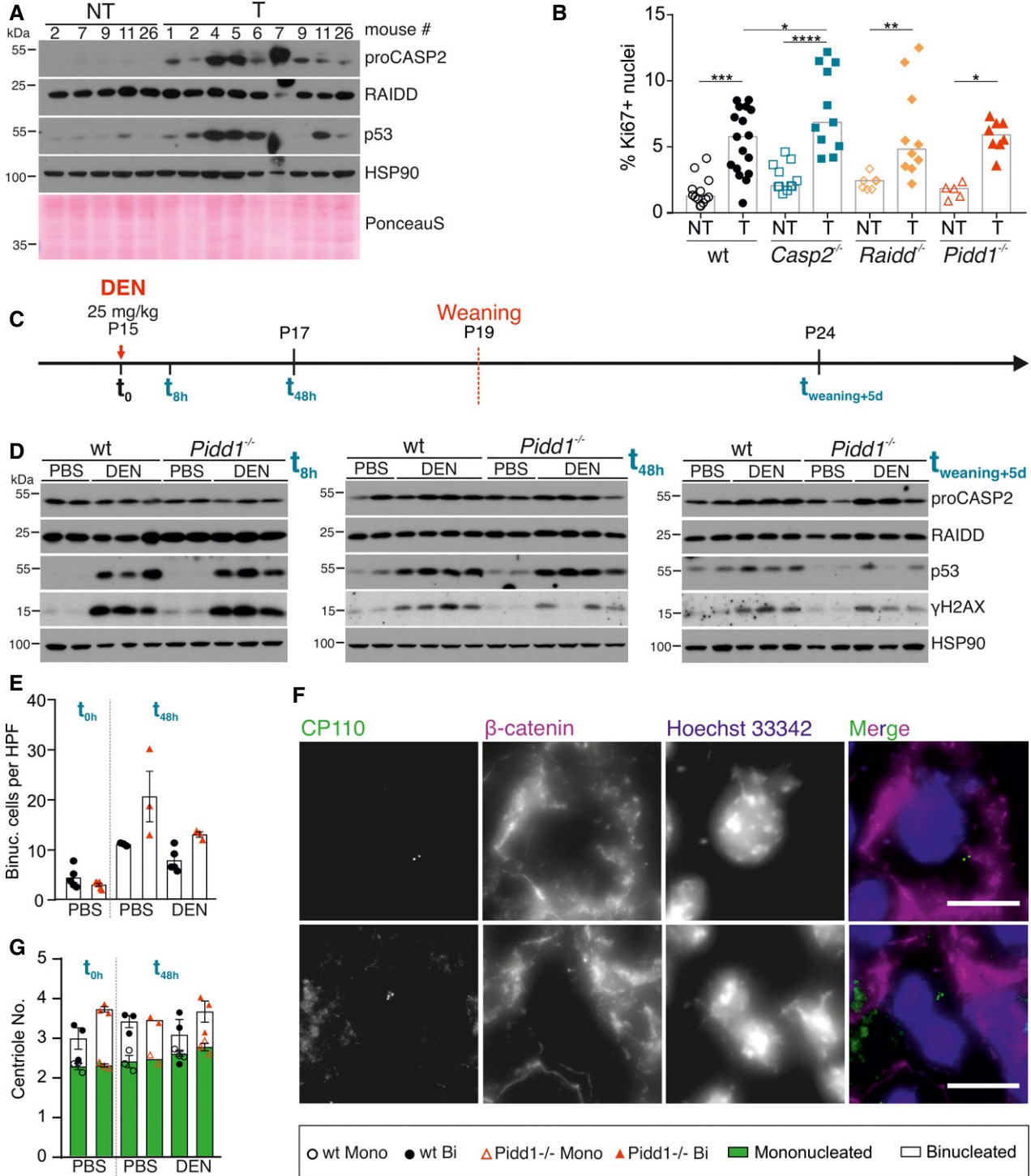

**Figure 2.**

**Figure 2.  DEN-induced liver tumors upregulate caspase-2 expression without a direct effect of DEN on PIDDosome activation.**

A   Immunoblots of wt non-tumorous (NT) and tumor (T) tissue lysates were probed for CASP2, RAIDD, and p53. HSP90 and PonceauS protein staining served as loading controls.

B   Proliferation of tumor tissue was determined by the percentage of Ki67-positive nuclei analyzed by flow cytometry. T: wt $n = 17$, $Casp2^{-/-}$ $n = 11$, $Raidd^{-/-}$ $n = 10$, $Pidd1^{-/-}$ $n = 8$; NT: wt $n = 12$, $Casp2^{-/-}$ $n = 11$, $Raidd^{-/-}$ $n = 7$, $Pidd1^{-/-}$ $n = 5$. N-numbers refer to biological replicates.

C   Experimental outline: 15-day-old (P15) female and male wt and $Pidd1^{-/-}$ mice were injected with DEN (25 mg/kg) or PBS at $t$(0). The livers were analyzed after $t$(8 h) and $t$(48 h), or the mice were weaned at P19 to induce polyploidization and livers harvested 5 days later $t$(weaning + 5 d). $t$(8 h) PBS: wt $n = 5$, $Pidd1^{-/-}$ $n = 5$; DEN: wt $n = 3$, $Pidd1^{-/-}$ $n = 3$. $t$(48 h) PBS: wt $n = 3$, $Pidd1^{-/-}$ $n = 3$; DEN: wt $n = 5$, $Pidd1^{-/-}$ $n = 3$. $t$(weaning + 5 d) PBS: wt $n = 3$, $Pidd1^{-/-}$ $n = 3$; DEN: wt $n = 3$, $Pidd1^{-/-}$ $n = 3$.

D   Immunoblots of the experiment described in (C) at $t$(8 h), $t$(48 h), and $t$(weaning + 5 d) for CASP2, RAIDD, p53, γH2AX, and HSP90 expression as loading control.

E   The number of binucleated hepatocytes per field was counted on H&E-stained paraffin sections of PBS treated wt and $Pidd1^{-/-}$ mice at $t$(0 h) and at $t$(48 h) after PBS or DEN treatment. N-numbers refer to biological replicates.

F   Immunofluorescence of the same livers shown in (E) performed for the distal centriolar protein CP110 to stain centrioles, β-catenin for cell borders, and DAPI to stain DNA. Representative images of a mononucleated hepatocyte with 2 centrioles and a binucleated hepatocyte with 4 centrioles are shown.

G   Centriole number was quantified with respect to the number of nuclei per cell in wt and $Pidd1^{-/-}$ livers at the time points described in (E); $n = 3$ biological replicates.

Data information: Data in (B) are represented as median, (E, G) as mean ± SEM, statistical significance was determined by one-way ANOVA with multiplicity correction (Sidak–Holm); *$P \leq 0.05$, **$P \leq 0.01$, ***$P \leq 0.001$, ****$P \leq 0.0001$. Scale bar size in (F) represents 10 μm.
Source data are available online for this figure.

Immunofluorescence of β-catenin was used to discriminate cell borders. Clearly, centriole number corresponded to the number of nuclei per cell in both genotypes and did not increase upon DNA damage (Fig 2F and G). Moreover, differences in the level of the enzyme Cyp2E1, needed to convert DEN into a DNA-alkylating intermediate, could also be excluded (Fig EV2E). Thus, we conclude that the PIDDosome is activated only after weaning to control the degree of polyploidization, but not upon DEN-induced DNA damage.

### Hepatocyte ploidy defines the tumor risk independent of caspase-2 and the PIDDosome

Based on the above, we reasoned that the unexpected reduction in liver tumors observed in mice lacking essential PIDDosome components might be linked to its function in ploidy restriction. As DEN-induced DNA damage and compensatory proliferation occur mainly in diploid cells after DEN injection, tumors may arise from a low ploidy state and hence the ploidy increase caused by PIDDosome deficiency at the time of weaning may become protective. To test this hypothesis, we assessed the ploidy state of murine DEN-induced tumor tissue (T) and matched non-tumorous liver tissue (NT). Remarkably, in tumor tissue the numbers of binucleated and hence clearly polyploid cells per field were significantly reduced and comparable in all tumors across genotypes. In contrast, PIDDosome-deficient non-tumorous tissue (NT) tended to harbor more binucleated hepatocytes when compared to wt livers (Fig 3A and B), consistent with our previous findings (Sladky *et al*, 2020). Moreover, we measured the ploidy of nuclei isolated from non-tumorous or tumor tissue which were stained with propidium iodide to assess DNA content by flow cytometry (Fig EV3A and B). Similar to binucleation, tumor cell nuclear ploidy, here exemplified by the percentage of octaploid nuclei, frequently found in healthy tissue (Celton-Morizur, 2010), was clearly decreased in tumor tissue of $Casp2^{-/-}$ and comparable to that found in wt mice. The same trend was observed for $Pidd1^{-/-}$ tumors. Curiously, $Raidd^{-/-}$ tumors did not show this effect (Fig 3C and D). Yet, all tumors in all genotypes tested showed lower levels of polyploidy, independent of the basal ploidy state of the surrounding parenchyma (Fig 3A). This supports the idea that HCC initiates from cells with low ploidy levels which

may have a higher risk of loss of heterozygosity (LOH) of certain tumor suppressors, and thus, are more likely to transform.

### Aneuploidy in HCC is linked to basal tumor cell ploidy but not limited by caspase-2

In the healthy liver the degree of aneuploidy increases with the polyploidy state of hepatocytes (Sladky *et al*, 2020). Of note, caspase-2 was reported to eliminate aneuploid cancer cells *in vitro* and possibly also in patients with colorectal cancer (Dawar *et al*, 2016; López-García *et al*, 2017). Therefore, we wanted to determine the degree of copy number variation (CNV) in wt and $Casp2^{-/-}$ liver tumors to assess the impact of PIDDosome loss on genomic stability and hence tumor evolution after transformation. To this end, we subjected diploid and tetraploid nuclei isolated from wt or $Casp2^{-/-}$ tumors to whole genome sequencing. Per tumor and ploidy state, 30 cells were pooled and subsequently sequenced to determine CNVs. Analysis of these mini-bulks allows conclusions on overall aneuploidy but also intra-tumor heterogeneity as reflected by non-integer copy number states (Fig 4A). In line with previous results in healthy liver (Sladky *et al*, 2020), the degree of CNV rises with the basal ploidy state but this phenomenon was independent of caspase-2 (Fig 4A and B), and hence, presumably, the entire PIDDosome, as well as PIDDosome-induced p53 activity. Together, this excludes a direct role for caspase-2 in aneuploidy tolerance, at least in hepatocellular carcinoma.

### Caspase-2 and PIDD1 are upregulated in human HCC and correlate with poor prognosis

Since we observed increased levels of caspase-2 in murine tumor samples, we next tested the levels of each PIDDosome component in human HCC tumors and matched non-tumorous hepatic tissue. Similar to the results obtained in mice, we found caspase-2 and PIDD1 protein to be clearly elevated in human tumor tissue compared to matched non-tumorous liver. Like in mice, RAIDD protein levels were comparable between tumor and healthy tissue (Figs 5A and EV4A). As reported before, cytokinesis failure caused by inhibition of Aurora B kinase activates the PIDDosome pathway

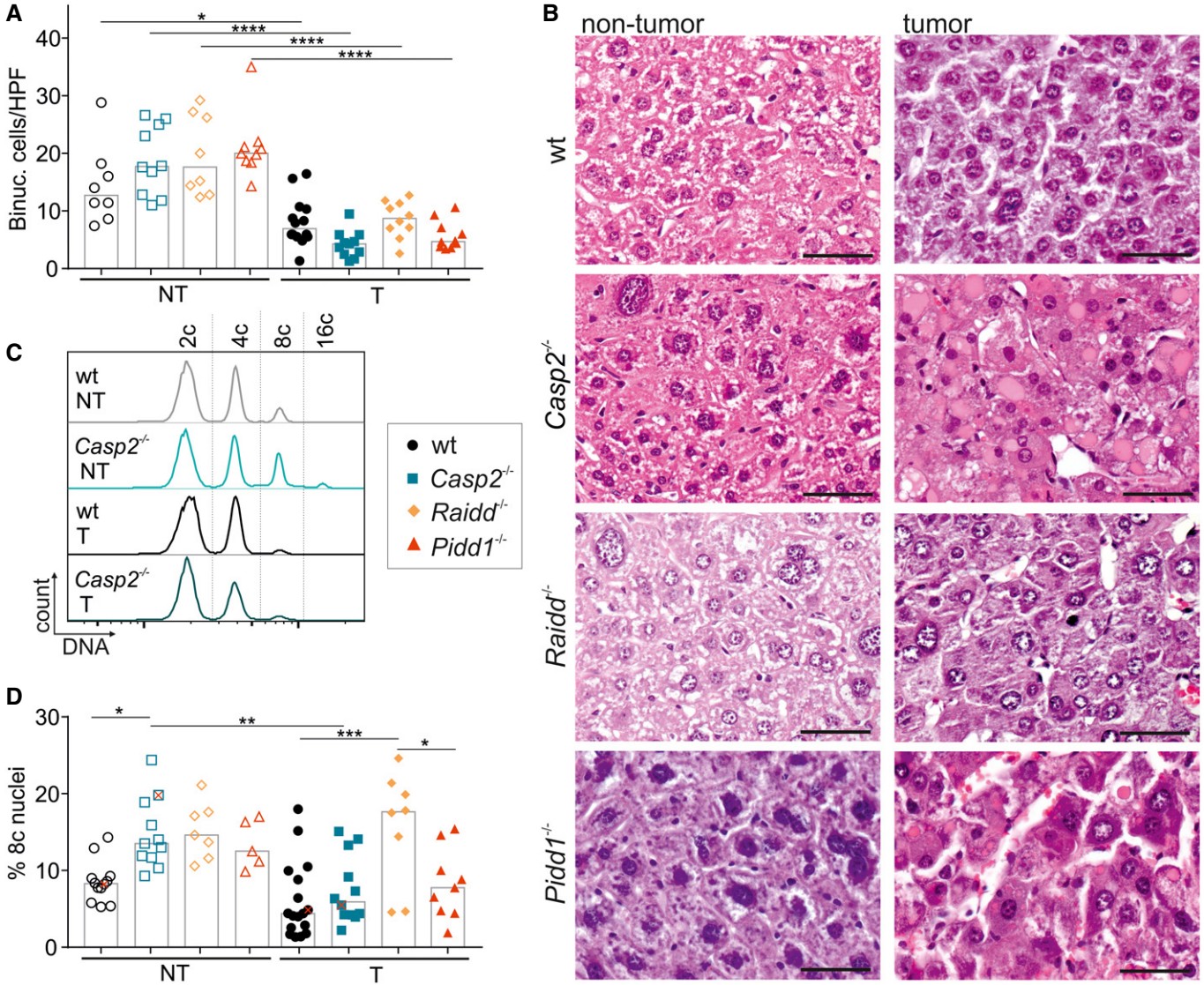

**Figure 3. Nuclear ploidy and binucleation are reduced in HCC tumor tissue.**

A, B (A) Binucleation was assessed blinded in H&E-stained sections of murine non-tumor (NT) and tumor tissue (T) also shown in (B). T: wt *n* = 14, *Casp2*$^{-/-}$ *n* = 11, *Raidd*$^{-/-}$ *n* = 10, *Pidd1*$^{-/-}$ *n* = 10; NT: wt *n* = 8, *Casp2*$^{-/-}$ *n* = 10, *Raidd*$^{-/-}$ *n* = 8, *Pidd1*$^{-/-}$ *n* = 9. *N*-numbers refer to biological replicates.

C, D (C) Representative histograms of nuclei stained for DNA, which were isolated from non-tumor and tumor tissue of wt and *Casp2*$^{-/-}$ mice. These animals are marked as red "x" in the quantification shown in (D) where the degree of polyploidy is reflected as percentage of octaploid nuclei (T: wt *n* = 17, *Casp2*$^{-/-}$ *n* = 11, *Raidd*$^{-/-}$ *n* = 10, *Pidd1*$^{-/-}$ *n* = 8; NT: wt *n* = 12, *Casp2*$^{-/-}$ *n* = 11, *Raidd*$^{-/-}$ *n* = 7, *Pidd1*$^{-/-}$ *n* = 5). *N*-numbers refer to biological replicates.

Data information: Data are represented as median, statistical significance was determined by one-way ANOVA with multiplicity correction (Sidak–Holm); *$P \leq 0.05$, **$P \leq 0.01$, ***$P \leq 0.001$, ****$P \leq 0.0001$; scale bar represents 100 μm.

in human HCC cell lines (Fig EV4B) (Sladky *et al*, 2020). Based on this observation, we wondered if *CASP2* or *PIDD1* mRNA expression levels may have prognostic value in HCC. Probing the Provisional Liver Hepatocellular Carcinoma TCGA data set we found that *CASP2* and *PIDD1* transcript levels are significantly upregulated in human HCC across all disease stages. Remarkably, CASP2 mRNA levels were even higher in patients lacking functional p53. In contrast, *RAIDD* expression showed the opposite trend (Fig 5B). In this data set, high *CASP2* expression correlated with reduced disease-free survival. This correlation, however, was not seen for *PIDD1* and *RAIDD* expression (Fig 5C).

**CASP2 and PIDD1 expression correlates with that of markers of proliferation**

*CASP2* and *PIDD1* expression is linked to proliferation of primary hepatocytes as both are transcriptional targets of E2F transcription factors (Sladky *et al*, 2020). Thus, we suspected that the observed effects on expression and survival in HCC patients could be due to increased proliferation rates in these tumors which may even be increased further in the absence of p53. Consistently, expression levels of well-known proliferation markers such as Ki67 also correlated with a shorter recurrence-free survival, similar to the reduced

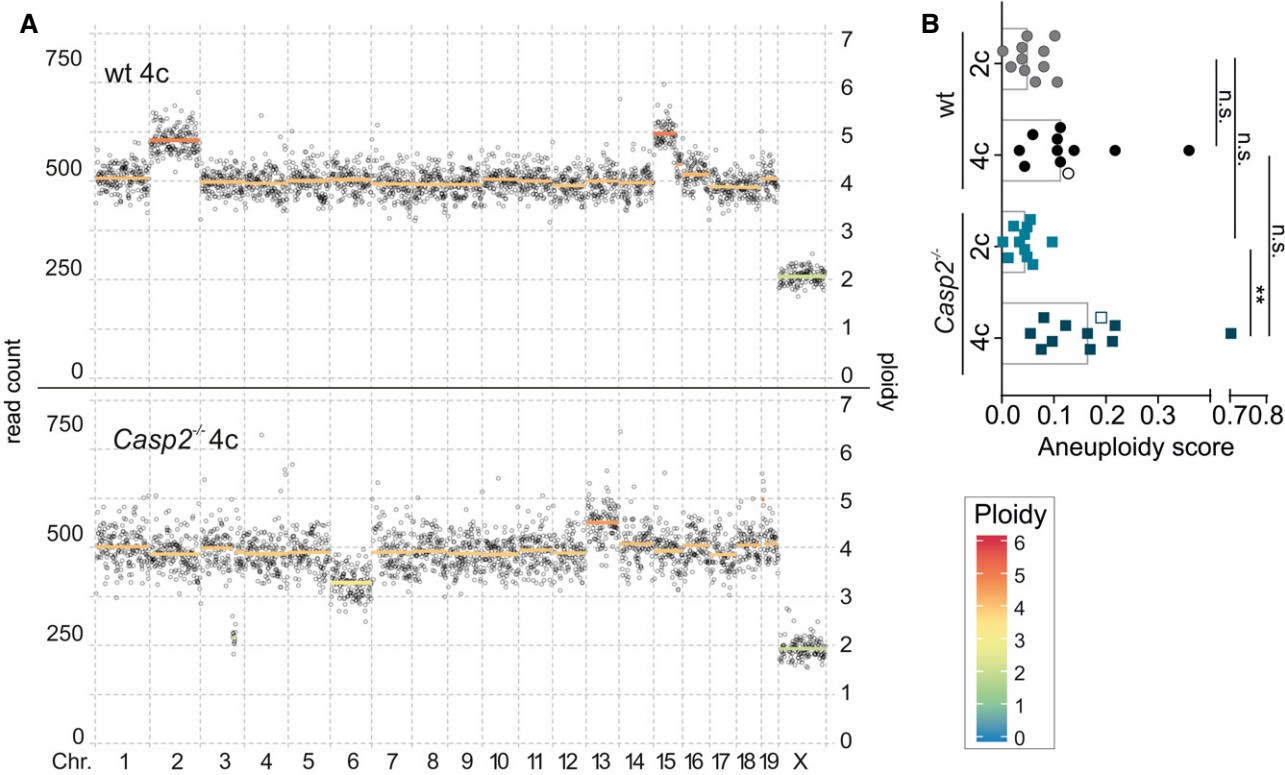

**Figure 4. Caspase-2 has no impact on the degree of aneuploidy in HCC.**

Copy number variations were determined using whole genome sequencing of 30 nuclei "mini-bulks" isolated from *Casp2*<sup>−/−</sup> or wt tumors (*n* = 11 per genotype). Diploid and tetraploid pools of tumor cell nuclei were analyzed separately.

A, B　(A) Representative dot plots show the degree of aneuploidy and heterogeneity and are quantified in (B, empty symbols). Data are represented as median, statistical significance was determined by one-way ANOVA with multiplicity correction (Sidak–Holm); *$P \leq 0.05$, **$P \leq 0.01$. *N*-numbers refer to biological replicates.

survival noted for patients with high *CASP2*-expressing HCC (Figs 5B and C, and EV4C). This effect is strongly affected by the mutational status of p53, indicating that these tumors are more aggressive. Of note, the disease-free survival of patients with wt p53 was not affected by the CASP2-expression level (Fig EV4D), suggesting again that higher mRNA levels may be linked solely to higher tumor-proliferation rates and are not predictive for disease outcome *per se*. Hence, we employed bioinformatics analyses of the same TCGA data set to assess which genes are co-regulated with the PIDDosome in HCC. Strikingly, the hallmark pathways associated with genes co-expressed with *CASP2* were mostly related to proliferation, including "G2/M checkpoint", "E2F targets", and "mitotic spindle". Although to a minor extent, the same pattern of co-expression was found for *PIDD1*. In contrast, *RAIDD* expression was not linked to proliferation markers but, unexpectedly, was found co-regulated with genes associated with lipid metabolism (Figs 6A and EV4E). Notably, the expression of *CASP2* and *PIDD1* positively correlates with other E2F target genes (Fig 6B). Moreover, *CASP2* and *PIDD1* expression showed a strong positive correlation with Ki67 transcript levels while the correlation for *RAIDD* mRNA with this proliferation marker was negative (Fig 6C–E).

Taken together, this suggests that the transcriptional upregulation in HCC tumors observed for *CASP2* and *PIDD1* is directly linked to the higher proliferation rates in the tumor while the expression of

RAIDD is uncoupled from the proliferative state. Thus, the impact of the PIDDosome on DEN-driven tumor formation in mice and the recurrence-free survival in HCC patients is most likely linked to the key role of the PIDDosome controlling cellular ploidy (and not to tumor cell proliferation *per se*).

### Tumor ploidy correlates with recurrence-free survival in HCC

Next, we asked whether the interrelation of polyploidy and murine HCC is also seen in liver cancer patients. Therefore, we assessed the cell density of tumor and matched non-tumorous tissue of 223 patients with histologically confirmed HCC by morphometric image analysis (Fig EV5B and C), as previously described (Rohr-Udilova *et al*, 2018). Cell density, or cells per field, is an indirect, reciprocal read-out for cell size and thus ploidy. For better data visualization, we used the reciprocal value of the cell density to calculate a "ploidy index" (1/cell density) for HCC tumor and non-tumorous tissue samples of each patient. Of note, the original cell density data were used to determine statistical significance and the "ploidy index" is used only for data representation. As shown before (Toyoda *et al*, 2005; Gentric *et al*, 2015; Bou-Nader *et al*, 2019), the etiology of the underlying liver diseases impacts on the degree of polyploidy. Non-tumorous tissue of patients with NASH or HBV infection was significantly

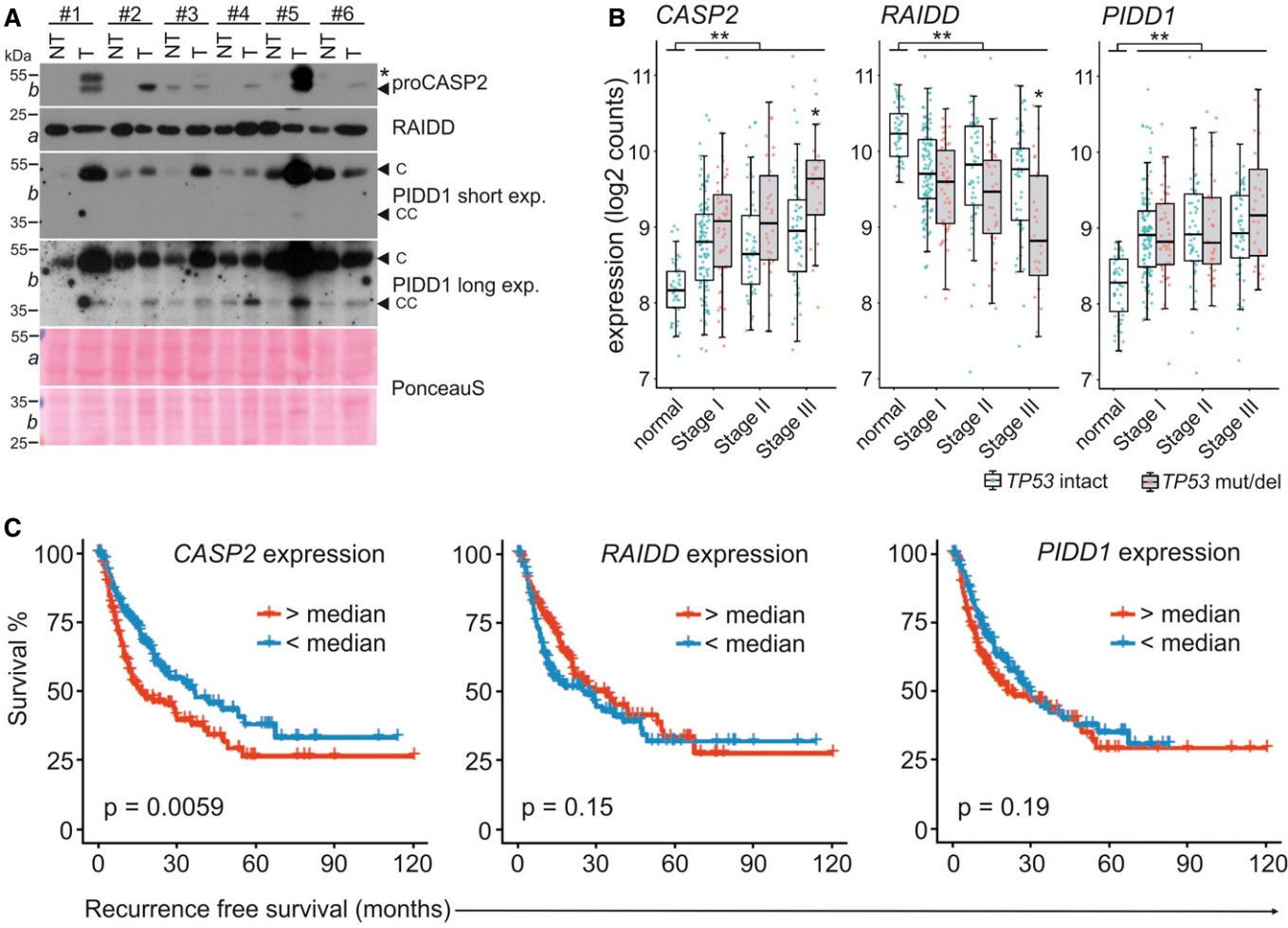

**Figure 5. Human HCC frequently upregulate CASP2 and PIDD1 expression.**

A   Matched patient non-tumor (NT) and tumor tissue (T) samples were analyzed by immunoblotting for CASP2, PIDD1, and RAIDD on membranes *a* and *b*. PonceauS protein staining serves as loading control. The asterisk refers to an unspecific band; "C" and "CC" indicate the functionally active fragments of PIDD1 generated by autoprocessing, PIDD-C and PIDD-CC. Densitometric quantification of these blots is shown in Fig EV4A.

B   mRNA expression analysis of the PIDDosome components in the TCGA Provisional LIHC data set across disease stages. Patients are divided based on the p53 mutation/deletion (mut/del) status. Statistical significance was determined using two-way ANOVA with Tukey's *post hoc* analysis. The central band represents the median, the boxes indicate the 25th and 75th percentile, the whiskers are the 5th and 95th percentiles, and statistical significance was defined as *$P \leq 0.05$, **$P \leq 0.01$.

C   The same data set was analyzed for recurrence-free survival based on high or low expression of *CASP2*, *RAIDD* or *PIDD1* divided at the median. Statistical significance was determined using a log-rank test.

Source data are available online for this figure.

more polyploid than HCV-infected liver (Fig 7A). Importantly, similar to our results obtained in mice and in line with previous studies (Toyoda *et al*, 2005; Gentric *et al*, 2015; Bou-Nader *et al*, 2019), we found that the ploidy in tumor tissue tended to be lower than in the surrounding liver. This decrease was highly significant for HCC patients with NASH (Fig 7A). To test whether the tumor ploidy affects disease outcome across the different disease etiologies, we pooled all available data on HCC tumor ploidy and divided the data according to the median HCC tumor cell density into two groups (Fig 7B). Remarkably, the group with low HCC tumor ploidy (high cell density) showed a significantly shorter recurrence-free survival (RFS). Strikingly, multivariate

analysis of age, sex, tumor size, vascular invasion, and tumor cell density clearly shows that the cell density represents an independent prognostic parameter for recurrence-free survival of HCC patients (Appendix Table S2).

Together, these findings in patients with HCC support the idea that low-ploid tumors are more proliferative and potentially more aggressive, increasing the risk of disease recurrence and resulting in impaired survival also after liver transplantation. As such, tumor cell density can serve as an independent prognostic marker, while *CASP2* mRNA expression serves as a surrogate of the proliferative capacity of tumor cells comparable in paucity to the well-established proliferation marker Ki67.

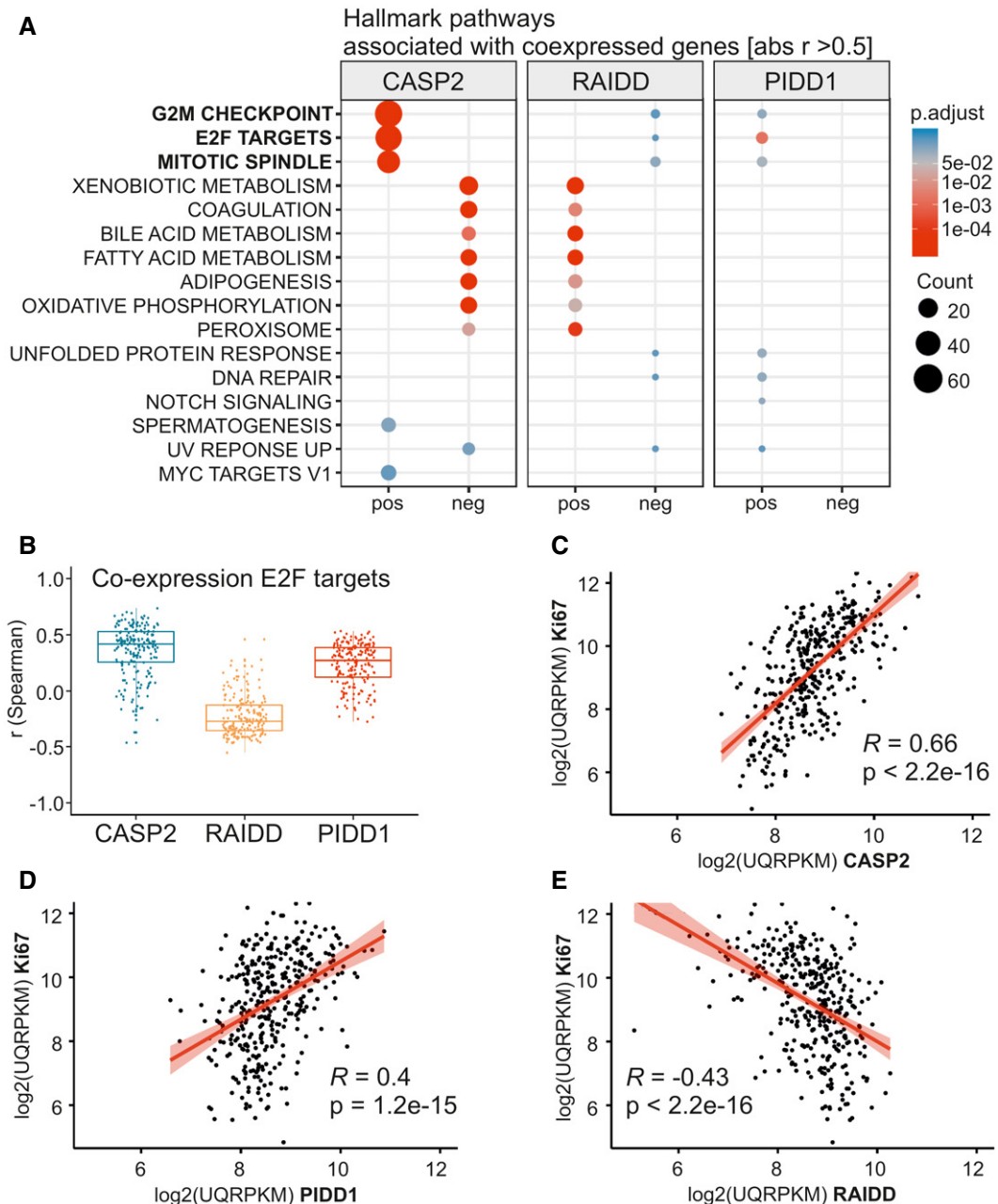

**Figure 6. Computational analysis of the TCGA Provisional LIHC data set for the expression of PIDDosome components reveals correlation with proliferation markers.**

A   Genes co-expressed with the PIDDosome components were examined with respect to the associated hallmark pathways taking positive (pos) and negative (neg) correlation into account.

B   E2F target genes expression was analyzed for correlation with expression of *CASP2*, *RAIDD*, and *PIDD1* using Spearman's correlation coefficient. The central band represents the median, the boxes indicate the 2nd and 3rd quartile, and the whiskers are 1.5 times the interquartile range ($n$ = 200 genes).

C–E   Correlation (Pearson) of *CASP2*, *RAIDD*, and *PIDD1* mRNA with transcript levels of *Ki67* in the TCGA data set.

# Discussion

We found that loss of PIDDosome-induced p53 function inhibits hepatocarcinogenesis, pointing toward an oncogenic feature of the most prominent tumor suppressor in the liver. This notion is supported by our observation that PIDDosome-deficient animals develop fewer tumors with an overall reduced tumor burden in DEN-induced HCC (Fig 1). Notably, DEN treatment induces compensatory proliferation in response to cell death that is the central pathogenic mechanism for disease onset in this model (Qiu

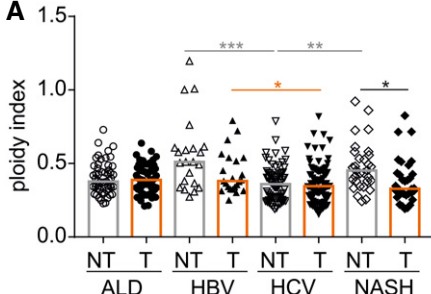

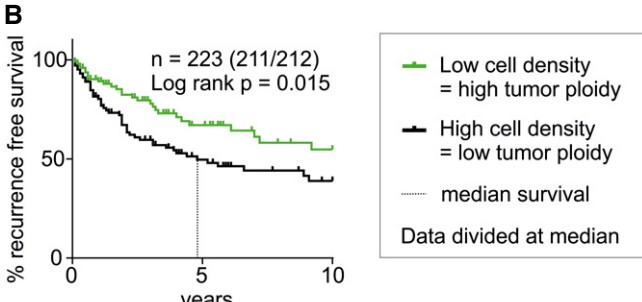

**Figure 7. The degree of polyploidy differs between etiologies, and low tumor ploidy correlates with reduced disease-free survival.**

A  The ploidy index is inferred as 1/cell density of tumorous (T) and non-tumorous (NT) tissue of HCC patients with different liver diseases. ALD—alcoholic liver disease (*n* = 65); HBV—hepatitis B virus infection (*n* = 24); HCV—hepatitis C virus infection (*n* = 85); NASH—non-alcoholic steatohepatitis (*n* = 33). Statistical significance was determined by one-way ANOVA with multiplicity correction (Sidak–Holm); *$P \leq 0.05$, **$P \leq 0.01$, ***$P \leq 0.001$. *n*-numbers refer to the number of patient samples analyzed.

B  All patients were grouped according to HCC tumor cell density as surrogate for tumor ploidy. Data were divided at the median, and exact patient numbers are shown in the graphs. Median recurrence-free survival (RFS) for a period of 10 years for patients with low tumor ploidy was 4.8 years. For patients with high tumor ploidy, the median RFS could not be defined as more than 50% of the patients survived without relapse. Patient characteristics are shown in Appendix Table S1. A log-rank test was used to determine statistical significance.

*et al*, 2011). Of note, similar findings have been made in a mouse model of irradiation-driven lymphoma where compensatory proliferation of hematopoietic stem and progenitor cells in response to radiation-induced apoptosis drives malignancy (Labi *et al*, 2010; Michalak *et al*, 2010). So, next to the induction of cell death, restriction of liver ploidy appears to be another example where p53 can become involuntarily oncogenic.

We found that *CASP2* and *PIDD1* expression is increased in human HCC as well as DEN-induced murine liver cancer (Figs 2, 5 and 6). As both genes are controlled by E2F transcription factors, their expression is clearly linked to proliferation (Sladky *et al*, 2020). Hence, the predictive value of *CASP2* expression on patient survival is most likely a secondary effect. As such, data correlating *CASP2* or *PIDD1* expression with patient survival need to be interpreted with caution as they most likely do not constitute markers of independent prognostic value.

Our analysis of RAIDD provides several interesting aspects. Aside from its function within the PIDDosome for liver ploidy control, it is not co-regulated with *CASP2* or *PIDD1* and RAIDD-deficient mice

developed the least number of tumors, albeit with the highest ploidy (Fig 1–5). It remains possible that the differences in phenotype observed in the absence of RAIDD are driven by a gain of function of PIDD1, that may then preferentially form alternative signaling complexes, such as the NEMO-PIDDosome (Sladky *et al*, 2017), or the recruitment of caspase-2 into alternative signaling platforms (Fava *et al*, 2012). Interestingly, however, RAIDD expression is actually negatively linked to proliferation but is co-expressed with metabolic enzymes, which points toward a potential role in liver metabolism. Based on its dual-adaptor domain structure, it seems to be predestined for functioning within alternative signaling complexes. Noteworthy here, RAIDD has originally been identified as a component in the TNF (tumor necrosis factor) receptor signaling complex (Duan & Dixit, 1997), which fuels liver carcinogenesis (Vucur *et al*, 2017).

Several recent reports describe the impact of liver cell ploidy on hepatocarcinogenesis (Zhang *et al*, 2018a,b; Lin *et al*, 2020). Our findings are well in line with this recently emphasized role of high ploidy as a barrier against HCC, as PIDDosome deficiency significantly increases hepatocyte ploidy (Sladky *et al*, 2020), a phenomenon that has been overlooked before in *Casp2* mutant mice (Shalini *et al*, 2016). Zhang *et al* (2018a, 2018b) provide solid evidence that polyploid hepatocytes harboring tumor suppressor genes in high copy number are protected from LOH and thus, malignant transformation. In accordance, mice harboring a liver-specific deletion of E2F8 and therefore mostly diploid hepatocytes develop spontaneous HCC at an early age (Kent *et al*, 2016). This emerging hypothesis that challenges the old dogma of polyploidy fostering aneuploidy and cancer is supported by the fact that both the cellular and nuclear ploidy levels of murine liver tumor tissue are reduced when compared to non-tumorous surrounding tissue (Fig 3). Strikingly, the degree of tumor ploidy was similar between wt and the otherwise highly polyploid *Casp2*$^{-/-}$ and *Pidd1*$^{-/-}$ livers (Fig 3). Equally noteworthy here, an increased occurrence of CNVs upon *Casp2* deletion or transcriptional repression, as noted in transformed mouse B cells or human colorectal carcinoma, respectively (Dawar *et al*, 2016; López-García *et al*, 2017), was neither found in the healthy liver (Sladky *et al*, 2020) nor in chemically induced HCC (Fig 4). Together, this suggests cell type-specific roles of caspase-2 in the restriction of aneuploidy. Admittedly, our mini-bulk sequencing approach may be geared to detect preferentially clonal aneuploidies and non-hepatocyte nuclei may skew our results in the 2C fraction analyzed. Yet, the number of infiltrating inflammatory cells in tumor tissue is modest and should not substantially alter these results, as it is also comparable across genotypes (Fig EV1).

In contrast to these previous studies documenting protective effects of higher hepatocyte ploidy at the time of genotoxic challenge (Kent *et al*, 2016; Zhang *et al*, 2018a,b; Lin *et al*, 2020), deficiency in caspase-2, PIDD1 or RAIDD leads to ploidy increases long after DNA damage has occurred and the DNA-damage response already ceases (Fig 2). At the time point of DEN injection (P15), most hepatocytes are still diploid (Fig 2), regardless of the genotype (Zhang *et al*, 2018b; Sladky *et al*, 2020). This suggests that polyploidy is not only protective at the time of mutagenesis, but can delay malignant transformation also if it occurs at a later stage, as shown in our study. This observation clearly challenges current thinking about the tumor-promoting role of polyploidization in cancer. A possible explanation may be that, for example, a tetraploid cell that had

duplicated a driver mutation at the time of weaning, now also harbors four copies of each tumor suppressor gene. As such, this gene duplication event shifts the balance in favor of the rheostat of tumor suppressors even at times long after the initial driver mutation has occurred. This may explain why tumors arise from hepatocytes of low ploidy, independent of the overall liver ploidy. As this low-ploid fraction of hepatocytes is substantially reduced in the absence of the PIDDosome, the population at risk for transformation is simply smaller.

Of note, most studies on the interrelation of ploidy and HCC were performed in mouse models where hepatocyte ploidy was modified by genetic manipulation of key cell cycle or cytokinesis regulators (Kent *et al*, 2016; Zhang *et al*, 2018a,b; Lin *et al*, 2020). However, enhanced polyploidy due to deregulated cell cycle proteins is not necessarily protective as mice overexpressing *CCNE1* show strong hyperpolyploidization but develop spontaneous liver cancer (Aziz *et al*, 2019). This finding challenges former results in support of the "ploidy-barrier" hypothesis. Yet, as PIDDosome deficiency increases hepatocyte ploidy without the need for direct manipulation of the cytokinesis or cell cycle machineries, our study suggests that the ploidy-barrier can prevent disease, until additional pro-proliferative signals force polyploid cells to repeatedly re-enter the cell cycle, e.g., upon inflammation.

Based on the above, it would be interesting to see whether the degree of ploidy in the human liver inversely correlates with HCC frequency. However, as HCC in humans is usually associated with an underlying inflammatory disorder that often changes hepatocyte ploidy, such a relationship will not easily be established (Toyoda *et al*, 2005; Gentric *et al*, 2015; Bou-Nader *et al*, 2019). Yet, we found a clear correlation between HCC tumor cell density and recurrence-free survival in our patient cohort (Fig 7). Automated morphometric cell density analysis as surrogate for the degree of polyploidy clearly showed that low HCC tumor ploidy is associated with reduced survival and could serve as independent prognostic parameter. However, experimental validation of the robustness of tumor cell density as a measure of cellular ploidy in matched patient samples subjected to morphometric as well as flow cytometric analysis is still pending and will be subject of future investigations.

A recent study suggested mononuclear polyploidy as marker for HCC aggressiveness, as in this cohort patients with highly polyploid, and often p53-mutated tumors, have an increased risk for recurrence (Bou-Nader *et al*, 2019). Of note, the patient cohort investigated by Bou-Nader *et al* comprised substantially fewer patients (75 vs. 223) and also the types of underlying diseases differ between our studies. Together, these differences might explain why our results, at first sight, appear to contrast the findings by Bou-Nader and colleagues. Clearly, p53 loss increases polyploidy tolerance but most certainly increases tumor aggressiveness and recurrence rates also by other means. Unfortunately, we do not have the corresponding information regarding p53 status for our patient cohort to test whether those carrying mutations would show reduced RFS within the high ploidy group. Further studies will have to assess whether healthy liver ploidy can indeed serve as an independent prognostic marker for the individual risk of HCC development. Morphometric analysis of routinely generated H&E-stained liver tissue biopsies, for example, from healthy donor livers used in transplant settings, could readily be used to collect data on hepatocyte ploidy in the healthy liver for prospective studies.

Taken together, we conclude that the PIDDosome indirectly impacts on chemically induced hepatocarcinogenesis by limiting hepatocyte polyploidization during postnatal liver development. Multiple lines of evidence suggest that caspase-2 is a promising therapeutic target (Sladky & Villunger, 2020). Here, we clearly demonstrate that caspase-2-mediated ploidy control impacts on HCC initiation. In addition, caspase-2 has been reported to accelerate NAFLD and NASH by induction of *de novo* lipogenesis, which can culminate in HCC (Machado *et al*, 2015; Kim *et al*, 2018; Anstee *et al*, 2019). Therefore, inhibition of caspase-2 for treatment of liver disease could prevent tumorigenesis both by elevated polyploidization and preventing the progression of hepatic steatosis.

# Materials and Methods

### DEN-induced HCC

All animal experiments were approved by the Austrian Bundesministerium für Bildung, Wissenschaft und Forschung (Tierversuchsgesetz 2012; https://www.ris.bka.gv.at/Dokumente/BgblAuth/BGBLA_ 2012_I_114/BGBLA_2012_I_114.pdfsig, BGBl I Nr. 114/2012, project number 66.011/0108/-WF/V/3b/2015). Generation and genotyping of $Casp2^{-/-}$, $Raidd^{-/-}$, and $Pidd1^{-/-}$ mice were previously described (O'Reilly *et al*, 2002; Berube *et al*, 2005; Manzl *et al*, 2009). All mice used were maintained on a C57BL/6N background under standard housing and enrichment conditions with a 12-h/12-h light/dark cycle. 15-day-old male C57BL/6N wild-type ($n = 22$), $Casp2^{-/-}$ ($n = 11$), $Raidd^{-/-}$ ($n = 14$), and $Pidd1^{-/-}$ ($n = 15$) mice were intraperitoneally injected during daytime with 25 mg/kg body mass DEN (diethylnitrosamine; Sigma-Aldrich, St. Louis, MO, Cat# N0258, CAS# 55-18-5) diluted in PBS and sacrificed for organ harvesting 10 months later. The health status of the mice was monitored visually and by monthly weight measurement. Tumor load was assessed by counting the number of tumor nodules on the liver surface, by measuring the size of surfaces tumors using a caliper, and by weighting the livers including tumors. Tissue samples of liver tumors and non-tumorous liver tissue were collected and either fixed in 4% paraformaldehyde or snap-frozen in liquid nitrogen. Blood was collected at the time of harvest. Serum levels of aspartate aminotransferase (AST), alanine aminotransferase (ALT), total bilirubin, urea, cholesterol, and triglycerides in the murine HCC models were measured enzymatically (Cobas analyzer, Roche Diagnostics, Mannheim, Germany) as by manufacturer's instruction. For short-term analyses, 15-day-old wt and $Pidd1^{-/-}$ mice were injected with PBS or DEN, as above. Mice were sacrificed after 8 h or 48 h, or were weaned at the age of 19 days and sacrificed 5 days later (experiment outline Fig 2C). Livers were flushed with PBS to remove blood and non-liver cells. Tissue was snap-frozen, fixed in 4% paraformaldehyde, or embedded and frozen in O.C.T. compound (Sakura Europe, Staufen, Germany, Cat# SA62550-01).

### Isolation of human HCC samples and non-tumorous liver tissue

Tumors and surrounding non-tumorous tissue samples of patients with histologically confirmed HCC undergoing orthotopic liver transplantation were collected at the Vienna General Hospital, Austria. The use of these biopsies for scientific research was approved by the ethics committee at the Medical University of

Vienna (#2033/2017), and patients gave informed consent. For the morphometric analyses, HCC tumor tissue and corresponding adjacent non-tumorous liver tissue from 223 patients were evaluated. Clinical data of the patients are summarized in Appendix Table S1. Clinical data analysis was performed in accordance with the guidelines of the local Ethics Committee at the Medical University of Vienna, and all patients gave written informed consent.

## Cell line experiments

SNU182 cells were obtained from the lab of Michael Trauner, Medical University of Vienna, Austria. Cells were cultured 37°C with 5% $CO_2$ in RPMI 1640 (Sigma-Aldrich, St. Louis, MO, Cat# FG1385) supplemented with 10% FBS (Cat# A15-151) and Penicillin/Streptomycin (Cat# P11-010, both PAA Laboratories, Pasching, Austria) and cultures regularly tested for mycoplasma contamination. Treatment with 1 or 2 μM ZM447439 (Selleck Chemicals, Houston, TX, Cat# S1103) for 48h was started 24h after seeding. Cells were harvested by trypsinization.

## Histopathological examination and immunohistochemistry

Histopathological analysis of H&E-stained 4-μm sections of paraffin-embedded liver was performed by board-certified veterinary pathologists, Laura Bongiovanni and Alain de Bruin, Dutch Molecular Pathology Centre, Utrecht University, NL. Number and approximate size of nodules, tumor malignancy, inflammatory infiltration, tumor necrosis, and binucleation were determined. Following deparaffinization and rehydration in xylol and ethanol, the tissue sections of non-tumorous liver and tumor tissue were treated 10 mM citrate buffer (pH 6.0) for antigen retrieval. Blocking was performed in peroxidase solution (1% $H_2O_2$/TBS), and sections were stained consecutively with primary (Rabbit monoclonal anti Ki67 (D3B5), Cell Signaling Technology, Danvers, MA, Cat# CS12202) and secondary (Biotin-SP-goat anti-rabbit AffiniPure, Jackson ImmunoResearch Laboratories, West Grove, PA, Cat# 111-065-144) antibody diluted in 5% goat serum/TBS. VECTASTAIN ABC Kit (Vector Laboratories, Burlingame, CA, Cat# PK-6100) and ImmPACT DAB (Vector Laboratories, Burlingame, CA, Cat# SK-4105) were used for amplification and development. The immune-stained sections were counterstained with hematoxylin, dehydrated, and mounted. Images were acquired on an Olympus BX45 microscope using cell^B software (version 2.7, Olympus soft imaging solutions, Shinjuku, Japan). Analysis of slides and images was performed blinded.

## Mini-bulk whole genome sequencing and data analysis

Isolation and sequencing of tumor and liver tissue nuclei was performed as described before (van den Bos et al, 2019). In brief, snap-frozen tissues were cut into small pieces and incubated in nuclei isolation buffer (10 mM Tris–HCl pH 8.0, 320 mM Sucrose, 5 mM $CaCl_2$, 3 mM $Mg(Ac)_2$, 0.1 mM EDTA, 1 mM DTT, 0.1% Triton X-100). Subsequently, nuclei were isolated from the tissue homogenate by gently passage through a 70-μm filter using a syringe plunger and pelleted by centrifugation. The nuclei pellet was resuspended in PBS containing 2% bovine serum albumin and 10 μg/ml Hoechst 33358 (Thermo Fisher Scientific, Waltham, MA,

Cat# H3569) and 10 μg/ml propidium iodide (Thermo Fisher Scientific, Waltham, MA, Cat# P1304MP). 30 nuclei per tumor were sorted into a single well for library preparation as described previously (van den Bos et al, 2019). Sequencing was performed using a NextSeq 500 (Illumina, San Diego, CA). The resulting sequencing data were aligned to the murine reference genome (GRCm38) using Bowtie2 (v2.2.4) (Langmead & Salzberg, 2012). To analyze copy number variation, the R package AneuFinder (v1.10.1) was used (Bakker et al, 2016). Following GC-correction, blacklisting of artifact-prone regions (extreme low or high coverage in control samples), and mappability checks, libraries were analyzed using the edivisive copy number calling algorithm using 1 Mb bin size, and modal copy number state according to the expected ploidy state and the breakpoint detection parameter was set to 0.9. The aneuploidy score of each library was calculated as average of the absolute deviation from the expected euploid copy number per bin.

## Isolation of nuclei from snap-frozen tissue

The procedure was adapted for liver and tumor tissue from Bergmann & Jovinge, 2012. Snap-frozen tissue was minced with a scalpel and transferred into tissue lysis buffer (0.32 M sucrose, 10 mM Tris–HCl pH 8, 5 mM $CaCl_2$, 5 mM MgAc, 2 mM EDTA, 0.5 mM EGTA, 1 mM DTT) and homogenized using an UltraTurrax probe homogenizer (IKA, Staufen, Germany) at 20,000 rpm for 12 s. To extract nuclei from intact cells, the homogenate was passed three times through a 20 g needle and remaining tissue pieces were removed by passages through 100- and 70-μm cell strainers. Nuclei were sedimented by centrifugation at 700 g for 10 min at 4°C and resuspended in 25 ml sucrose buffer (2.1 M sucrose, 10 mM Tris–HCl pH 8, 5 mM MgAc, 1 mM DTT). For high-density centrifugation in order to remove cell debris and enrich for nuclei, BSA-coated ultracentrifugation tubes prefilled with 10 ml sucrose buffer cushion were carefully overlaid with the dissolved nuclei pellet and centrifuged for 60 min at 13,000 g (Beckman, Brea, CA, Avanti S-25, JA25.50 rotor). The supernatant was discarded, and the isolated nuclei were dissolved in 1 ml nuclei buffer (0.44 M sucrose, 10 mM Tris–HCl pH 7.4, 70 mM KCl, 10 mM $MgCl_2$).

## Flow cytometry

$3 \times 10^6$ nuclei in nuclei isolation buffer were centrifuged at 700 g for 10 min fixed by adding 100 μl BD Cytofix/Cytoperm solution (BD Biosciences, Franklin Lakes, NJ, Cat# 554714) while vortexing and incubated for 15 min on ice. After a washing step with 2% BSA in PBS, nuclei were resuspended in 100 μl Perm/Wash buffer (BD Biosciences, Franklin Lakes, NJ, Cat# 554714, prediluted 1:10 in A.D.) and incubated for 10 min at room temperature. Following a washing step with Perm/Wash, the nuclei were incubated with A488-conjugated antibody recognizing Ki67 (Alexa Fluor® 488 anti-mouse Ki67 Antibody (16A8), BioLegend, San Diego, CA, Cat# 652417) diluted 1:50 in Perm/Wash for 30 min at room temperature. Nuclei were washed with 2% BSA in PBS and resuspended with 1 μg/ml propidium iodide in PBS to stain DNA. The stained nuclei were directly analyzed by flow cytometry on a LSR Fortessa (BD).

## Immunofluorescence

O.C.T.-embedded liver cryo-sections of $n = 3$ wt or $Pidd1^{-/-}$ mice per time point and treatment were fixed in 100% methanol and stained for CP110 (rabbit polyclonal anti-CP110, Proteintech, Rosemont, IL, Cat#1278-1-AP) and ß-catenin (mouse monoclonal anti-β-catenin (14), BD Bioscience, Franklin Lakes, NJ, Cat# 610154). The secondary antibodies used were Alexa 488 goat anti-rabbit (A11031) and Alexa 546 goat anti-mouse (A11030) from Thermo Fisher Scientific, Waltham, MA. The nuclei were stained with Hoechst 33342 (Sigma-Aldrich, St. Louis, MO, CAS# 23491-52-3), and the sections were mounted in Mowiol. Images were acquired (2,048 × 2,048 pixels, bit depth: 16 bit) on a Zeiss Axiovert 200M microscope with an oil immersion objective (Ph3 Plan-Neofluar 100×/1.3 oil, 440481, Zeiss, Oberkochen, Germany) using the acquisition software VisiView 4.1.0.3 (Visitron Systems, Puchheim, Germany). ImageJ was used to adjust contrast and brightness of the images. Per mouse, 50 cells were analyzed. The number of nuclei and centrioles was counted manually in a blinded fashion.

## Protein isolation and immunoblotting

Snap-frozen tissue of mouse or human origin and cell line pellets were lysed in Ripa buffer (150 mM NaCl, 50 mM Tris, 1% NP-40, 0.5% sodium deoxycholate 1% SDS, 1 tablet EDTA-free protease inhibitors). The protein content was determined using Bradford's reagent (Bio-Rad, Hercules, CA, 500-0006), and 40–80 μg protein was used for Western blotting (AmershamTM HybondTM—ECL nitrocellulose membranes, GE Healthcare, Chicago, IL) with a wet-transfer system (Bio-Rad, Hercules, CA). Proteins of interest were detected using antibodies recognizing caspase-2 (rat monoclonal anti-Casp2 (11B4), Enzo Life Sciences, Farmingdale, NY, Cat# ALX-804-356; 1:1,000), RAIDD (rabbit polyclonal anti-RAIDD, Proteintech, Rosemont, IL, Cat# 10401-1-AP, 1:1,000), PIDD1 (mouse monoclonal anti-PIDD1 (Anto-1), Enzo Life Science, Cat# ALX-804-837, 1:500), p53 (Mouse monoclonal anti-p53 (1C12), Cell Signaling Technology, Danvers, MA, Cat# CS2524), Phospho-Histone H2A.X (Ser139, Cell Signaling Technology, Danvers, MA, Cat# CS2577), MDM2 (mouse monoclonal anti-MDM2 (IF2), Thermo Fisher Scientific, Waltham, MA, Cat# MA1113), and HSP90 (mouse monoclonal anti-Hsp90, Santa Cruz, Dallas, TX, Cat# sc-13119, 1:10,000). HRP-conjugated secondary antibodies used for detection were diluted 1:5,000 (HRP-mouse anti-rat, Cell Signaling Technology, Danvers, MA, Cat# CS7077S; HRP-mouse anti-mouse, Dako, Agilent, Santa Clara, CA, Cat# P0616; HRP-mouse anti-rabbit, Dako, Cat# P0448). All antibodies used were verified with appropriate controls in the present and in previous publications (Fava et al, 2017; Sladky et al, 2020). Densitometric quantification was performed using ImageJ.

## RNA isolation and quantitative real-time PCR

Snap-frozen tissue was grinded in liquid nitrogen and dissolved in 500 μl TRIzol. RNA was isolated by adding chloroform to 1/5 $v/v$. Samples were mixed and incubated at room temperature. After centrifugation at 12,000 $g$ for 15 min at 4°C, the aqueous phase was transferred into fresh tubes and the equivalent volume of isopropanol was added, mixed, and incubated for 10 min at room temperature. Purified RNA was sedimented by centrifugation and washed with 75% ethanol. The dried pellets were resuspended in DEPC-treated A.D., and DNA was digested using RQ1 RNase-Free DNase kit according to manufacturer's instructions (Promega, Madison, WI, Cat#: M6101). Finally, RNA was precipitated again with Glyco-BlueTM Coprecipitant (Thermo Fisher Scientific, Waltham, MA, Cat#: AM9515). The RNA content was quantified with a NanoDrop 1000 Spectrophotometer (Thermo Fisher Scientific, Waltham, MA) and reverse transcribed into cDNA with iScript™ cDNA Synthesis Kit (Bio-Rad, Hercules, CA, Cat#: 1708891) as stated in the manufacturer's instructions. Quantitative real-time PCR was performed on a StepOnePlus System (Thermo Fisher Scientific, Waltham, MA). The fluorescence dye-based qRT–PCR using AceQ qPCR SYBR® Green Master Mix (Vazyme, Nanjing, China, Cat#: P410) was performed according to the manufacturer's protocol (10 μl 2 × AceQ qPCR SYBR® Green Master Mix, 0.4 μl Rox 1, 0.2 μl Primer Mix, 2 μl cDNA filled up with A.D. to 20 μl). The following primers were used at a final concentration of 100 nM for detection of Casp2, Pidd1, Raidd and Hprt expression: mCasp2 fw: TCTCACATGGTGTGGAAGGT, mCasp2 rev: AGGGGATTGTGTGTGGTTCT, mPidd1 fw: TGTTCTGCACAGCAACCTCC, mPidd1 rev: TGGGATATGTCTGGGGGACT, mRaidd fw: GCTTATCGGAAGAAATGGAAGCC, mRaidd rev: GGCCTGTGGTTTGAGCTTTG (Sladky et al, 2020), mHprt fw: GTCATGCCGACCCGCAGTC, and mHprt rev: GTCCTTCCATAATAGTCCATGAGGAATAAAC (Schoeler et al, 2019). mRNA expression of Cyp2E1, Prom1, and Thy1 was assessed using predesigned primer assays (Integrated DNA Technologies, Inc, Coralville, IA, USA. Cyp2E1: Mm.PT.58.9617541; Prom1: Mm.PT.58.8851198; Thy1: Mm.PT.58.14029994). All qPCRs were performed in duplicates. PCR conditions were as follows: 95°C for 10 min, 40 cycles of (95°C for 15 s and 60°C for 60 s) and a melting curve with 0.3°C increment steps up to 95°C for 15 s. Results for the gene of interest were normalized to expression of housekeeping gene Hprt, and relative expression was calculated using the $\Delta\Delta C_t$ method.

## Data analysis

Statistical analyses were performed by Prism 7.0.0 (GraphPad Software, San Diego, CA). Unpaired Student's $t$-test was used for comparison of two groups, and one-way ANOVA with multiple comparison correction (Sidak–Holm) was employed if more than two groups were compared. All relevant comparisons were made, and non-significant results were not indicated in the figures. Significance levels are stated in the respective figure legends.

Gene expression data of hepatocellular carcinoma samples collected for The Cancer Genome Atlas (TCGA) project (https://www.cancer.gov/tcga) were downloaded from the GDAC Firehose website (http://firebrowse.org; 28/01/2018 release) as upper quartile normalized RPKM (UQRPKM) values provided by RSEM. Clinical information on patients, including disease-free survival, was learned from the revised data set of Liu et al (2018) extracted recently from the TCGA database. For each patient, the first primary solid tumor tissue sample in alphabetical order of sample barcodes was taken into account. Kaplan–Meier curves and log-rank statistics to compare prognosis in groups were analyzed using the survminer R package (https://CRAN.R-project.org/package = survminer). Information on the p53 mutation status was downloaded from cBioPortal (cbioportal.org).

Pairwise Pearson's correlation coefficients of gene expression in hepatocellular carcinoma samples were downloaded from the

cBioPortal for *Ki67*, *E2F1*, *CCNA2*, *PIDD1*, *RAIDD* (*CRADD*), and *CASP2* (cbioportal.org) (Gao *et al*, 2013). Genes with a Pearson's r value greater than 0.5 or smaller than −0.5 were considered as co-regulated with the target genes. The resulting gene lists were subjected to enrichment analysis with the *enricher* function of clusterProfiler in R (Yu *et al*, 2012). As an alternative approach, Gene Set Enrichment Analysis with all genes ranked by the *r* value was also carried out using clusterProfiler. Signature gene sets including transcription factor targets and GO terms were downloaded from the MSigDB database (http://software.broadinstitute.org/gsea/msigdb) using the msigdbr package (Subramanian *et al*, 2005).

### Cell density measurement

To assess cell density tissue arrays as shown in Rohr-Udilova *et al* (2018) were used. Nuclear staining was performed by hematoxylin. Slide images were digitized using a Pannoramic Midi Slide Scanner (3Dhistech, Budapest, Hungary). Morphometric characteristics of single nuclei were evaluated by tissue morphometric analysis of the digitized slide images using the Tissue Studio® software (Definiens, Munich, Germany). On average, $8{,}710 \pm 3{,}425$ cells in tumor and matched non-tumorous tissue sections were analyzed per patient. The tissue sections were previously stained for tryptase to detect mast cells (Rohr-Udilova *et al*, 2018). Negatively stained cells were considered for further morphometric analysis. As shown in Fig EV5B and C, the software allows the detection of nuclei and cell borders to measure the cell area. To test the impact of nuclear circularity images of 39 HCC patients were reanalyzed considering only nuclei with a circularity-index greater than 7 out of 10 for survival analysis (Fig EV5A). As no difference was seen, all nuclei stained negatively for tryptase were used for further analysis to assess cell density (Fig EV5B and C). Recurrence-free survival was depicted by Kaplan–Meier curves and compared between patient groups by log-rank Mantel–Cox test using Prism 7.0.0, GraphPad Software.

### Multivariate analysis

The contribution of cell density on the recurrence-free survival of patients undergoing transplantation for HCC was evaluated. Recurrence and/or death were considered as event while patients were censored at last clinical contact. Parameters associated with recurrence-free survival on univariate analysis were tested for their independent prognostic impact by Cox regression and stepwise backward elimination. In total, the analysis included $n = 172$ patients, as not all data were available for the 223 patients included in the cell density analysis. Calculations were performed using IBM SPSS Statistics 25 Software.

## Data availability

The single-cell sequencing data set generated in this study is available on the European Nucleotide Archive (https://www.ebi.ac.uk/ena/browser/home); Accession number: PRJEB3593.

**Expanded View** for this article is available online.

## Acknowledgements

We are grateful to K. Rossi, I. Gaggl, J. Heppke, C. Soratroi, and A. Beierfuß for excellent technical assistance or animal care. We also thank M. Trauner, A. Strasser, and T. Mak for sharing cell lines, mouse models, and reagents as well as A. Curinha, S. Sprung, J. Haybäck, M. Bergmann, and G.F. Vogel for fruitful discussion and support with histology and microscopy. This work was supported by the FWF-funded Doctoral College "Molecular Cell Biology and Oncology" (W1101) and the ERC-AdG "POLICE" (#787171). Artwork was created using Biorender.com.

## Author contributions

VCS conducted and designed experiments, analyzed data, prepared figures, and wrote manuscript. KK, VZB, and AC performed experiments. TGS and BW analyzed TCGA liver cancer data. LB and AB performed histological analyses. HB, DCJS, and FF conducted single-cell whole genome sequencing and analyzed the results. TS and HS analyzed serum parameters. TR and GS analyzed HCC data sets. TR enabled access to human tissue specimens. GT, KT, and MP performed morphometric image analyses. NR-U performed morphometric analysis of human tissue sections and analyzed HCC data sets. AV designed research, analyzed data, wrote manuscript, and conceived study.

## Conflict of interest

The authors declare that they have no conflict of interest.

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
