## [Review Process File · EMBO Reports]

PIDDosome-induced p53-dependent ploidy restriction facilitates hepatocarcinogenesis

Valentina Sladky, Katja Knapp, Tamas Szabo, Vincent Braun, Laura Bongiovanni, Hilda van den Bos, Diana Spierings, Bart Westendorp, Ana Curinha, Tatjana Stojakovic, Hubert Scharnagl, Gerald Timelthaler, Kaoru Tsuchia, Matthias Pinter, Georg Semmler, Floris Fojjer, Alain de Bruin, Thomas Reiberger, Nataliya Rohr-Udilova, and Andreas Villunger

DOI: 10.15252/embr.202050893

Corresponding author(s): Andreas Villunger (andreas.villunger@i-med.ac.at)

Review Timeline:

Submission Date:	13th May 20
Editorial Decision:	10th Jun 20
Revision Received:	1st Sep 20
Editorial Decision:	28th Sep 20
Revision Received:	13th Oct 20
Accepted:	14th Oct 20

Editor: Achim Breiling

Transaction Report:

Dear Andreas,

Thank you for the transfer of your research manuscript to EMBO reports. We have now received reports from the three referees that were asked to evaluate your study, which can be found at the end of this email.

As you will see, all referees think that the findings are of interest, but they also have several comments, concerns and suggestions, indicating that a major revision of the manuscript is necessary to allow publication in EMBO reports. As the reports are below, and I think all points need to be addressed, I will not detail them here.

Given the constructive referee comments, we would like to invite you to revise your manuscript with the understanding that all referee concerns must be addressed in the revised manuscript and in a detailed point-by-point response. Acceptance of your manuscript will depend on a positive outcome of a second round of review. It is EMBO reports policy to allow a single round of revision only and acceptance of the manuscript will therefore depend on the completeness of your responses included in the next, final version of the manuscript.

Revised manuscripts should be submitted within three months of a request for revision. We are aware that many laboratories cannot function at full efficiency during the current COVID-19/SARS-CoV-2 pandemic and we have therefore extended our 'scooping protection policy' to cover the period required for full revision. Please contact me to discuss the revision should you need additional time, and also if you see a paper with related content published elsewhere.

PLEASE NOTE that upon resubmission revised manuscripts are subjected to an initial quality control prior to exposition to re-review. Upon failure in the initial quality control, the manuscripts are sent back to the authors, which may lead to delays. Frequent reasons for such a failure are the lack of the data availability section (please see below) and the presence of statistics based on $n=2$ (the authors are then asked to present scatter plots or provide more data points).

- 1) a .docx formatted version of the final manuscript text (including legends for main figures, EV figures and tables), but without the figures included. Please make sure that changes are highlighted to be clearly visible. Figure legends should be compiled at the end of the manuscript text.
- 2) individual production quality figure files as .eps, .tif, .jpg (one file per figure), of main figures and EV figures. Please upload these as separate, individual files upon re-submission.

The Expanded View format, which will be displayed in the main HTML of the paper in a collapsible format, has replaced the Supplementary information. You can submit up to 5 images as Expanded View. Please follow the nomenclature Figure EV1, Figure EV2 etc. The figure legend for these should be included in the main manuscript document file in a section called Expanded View Figure Legends after the main Figure Legends section. Additional Supplementary material should be supplied as a single pdf file labeled Appendix. The Appendix should have page numbers and needs

to include a table of content on the first page (with page numbers) and legends for all content. Please follow the nomenclature Appendix Figure Sx, Appendix Table Sx etc. throughout the text, and also label the figures and tables according to this nomenclature.

For more details please refer to our guide to authors:

See also our guide for figure preparation:

http://wol-prod-cdn.literatumonline.com/pb-assets/embosite/EMBOPress_Figure_Guidelines_061115-1561436025777.pdf

4) a complete author checklist, which you can download from our author guidelines (<https://www.embopress.org/page/journal/14693178/authorguide>). Please insert page numbers in the checklist to indicate where the requested information can be found in the manuscript. The completed author checklist will also be part of the RPF.

Please also follow our guidelines for the use of living organisms, and the respective reporting guidelines: <http://www.embopress.org/page/journal/14693178/authorguide#livingorganisms>

5) that primary datasets produced in this study (e.g. RNA-seq, ChIP-seq and array data) are deposited in an appropriate public database. This is now mandatory (like the COI statement). If no primary datasets have been deposited in any database, please state this in this section (e.g. 'No primary datasets have been generated and deposited').

The accession numbers and database should be listed in a formal "Data Availability " section (placed after Materials & Methods) that follows the model below. Please note that the Data Availability Section is restricted to new primary data that are part of this study.

Data availability

6) We strongly encourage the publication of original source data with the aim of making primary data more accessible and transparent to the reader. The source data will be published in a separate source data file online along with the accepted manuscript and will be linked to the relevant figure. If you would like to use this opportunity, please submit the source data (for example scans of entire gels or blots, data points of graphs in an excel sheet, additional images, etc.) of your key experiments together with the revised manuscript. If you want to provide source data, please include size markers for scans of entire gels, label the scans with figure and panel number, and send one PDF file per figure.

8) Regarding data quantification and statistics, can you please specify, where applicable, the number "n" for how many independent experiments (biological replicates) were performed, the bars and error bars (e.g. SEM, SD) and the test used to calculate p-values in the respective figure legends. Please provide statistical testing where applicable, and also add a paragraph detailing this to the methods section. See: <http://www.embopress.org/page/journal/14693178/authorguide#statisticalanalysis>

Please note that all corresponding authors are required to supply an ORCID ID for their name upon submission of a revised manuscript. Please find instructions on how to link the ORCID ID to the account in our manuscript tracking system in our Author guidelines: <http://www.embopress.org/page/journal/14693178/authorguide#authorshipguidelines>

I look forward to seeing a revised version of your manuscript when it is ready. Please let me know if you have questions or comments regarding the revision.

Kind regards,

Achim

Achim Breiling
Editor
EMBO Reports

Referee #1:

The manuscript by Sladky et al. is an extension of their 2020 Developmental Cell paper that established a role for the PIDDosome in regulation of hepatic polyploidy. The current study examines the connection between PIDDosome, ploidy and HCC formation in the DEN mouse model and in patients. The authors clearly demonstrate a number of important findings with high

translational potential:

- * DEN-induced tumor formation in Casp2- and PIDDosome-deficient mice is reduced;
- * Casp2 and Pidd1 (lowly expressed in adults) are highly expressed in HCC in mice and humans;
- * HCC patients with low cell density (and presumably high ploidy) have improved recurrence-free survival.

Overall, this is a potentially very interesting paper. It is well written, experiments are high quality and use some novel approaches, statistics are appropriate. The topic is timely and relevant. There are number of issues that dampen my enthusiasm. Most significantly, it is unclear how PIDDosome affects tumorigenesis as the links between PIDDosome, ploidy and tumors are unclear. Specific concerns - major and minor - are outlined below and should be addressed.

MAJOR CONCERNS:

1. The authors compared tumorigenesis using the established DEN model and found that germline loss of Casp2, Pidd1 or Raidd resulted in formation of fewer tumors and smaller tumors, although LW/BW remained equivalent. Overall, analysis of the mice and tumors after 10 months is well done. There is a lot of literature describing the types of tumors generated in the DEN model. Does loss of Caspase 2 or PIDDosome components fundamentally alter tumorigenesis - in other words, does the spectrum of tumors in each of the KOs differ from WT?
2. Using the TCGA database, the authors show in Fig. 5 that CASP2 and PIDD1 expression correlates with HCC severity, and that patients with low CASP2 expression have modest, yet statistically significant, improvement in recurrence-free survival compared to patients with high CASP2 expression. Moreover, analysis showed that CASP2 and PIDD1 expression correlate with proliferation-associated gene signatures. These observations are important as they essentially replicate the findings in mice and strongly support a role for CASP2 and PIDD1 in HCC formation. My major concern with this section is the author's attempt to link PIDDosome expression with ploidy alterations to tumorigenesis. The concluding sentence on page 19 is especially confusing, "Thus, the impact of PIDDosome on hepatocarcinogenesis in mice and the recurrence-free survival in HCC patients is most likely not related to direct effects of the on tumor cell proliferation but rather linked to the key role of the PIDDosome controlling ploidy." The logic needs to be clarified or the conclusion needs to be revised.
3. Based on data in Fig. 7, the authors suggest the exciting idea that tumor cell density can be used as a marker for recurrence-free survival in HCC patients. Basically, low cell density (and, thus, high ploidy ... see minor comment #3 below) have longer recurrence-free survival. This is supported by mouse studies (e.g., Zhang, Dev Cell 2018; Wilkinson, Hepatology 2019; others). This finding contrasts with Bou-Nader, et al (Gut 2019) where patient HCC are enriched with mononuclear polyploid hepatocytes, especially those with p53 mutations. The authors address this issue in the Discussion, citing the increased patient "n" and the lack of p53 mutational status in the current study. Despite this reasonable argument, I'm left wondering which set of observations/conclusions is correct. Any extra data the authors can provide would be tremendously valuable.

MINOR CONCERNS:

1. The method to measure CNV in "mini bulks" from tumor tissue is a clever approach. I can see two limitations, which should be discussed. First, only clonal aneuploidy should be detected with this approach. Random CNV by any of the 30 cells may be below the limit of detection and, therefore, missed. Second, since the nuclei are isolated from tissue, any nuclei from non-parenchymal cells,

inflammatory cells, etc. would contaminate the prep, likely skewing the pool toward euploidy. This should not be an issue with the 4c cells but could affect the 2c cells. Despite these limitations, the CNV patterns between WT and Casp2-KO are similar, and I think the data are sound.

2. In Fig. 5A, the pattern of proCASP2 in NT and T tissue is easy to see, but it is harder to appreciate the pattern with PIDD2 expression, which is more heterogeneous. It would be helpful to quantify the data and display as a graph.

3. The approach in Fig. 7 to use cell density as readout for ploidy is potentially very useful. Unfortunately, this approach is never verified. The authors should compare the cell density approach with the gold standard method for determining ploidy - FACS analysis. I would suggest comparing ploidy in WT and one or more of the KO mice by both methods. Another useful analysis would be to determine the ploidy index in mouse NT and T tissues and compare with FACS data in Fig. 3.

4. After reading the Results surrounding Figure 3, I initially felt the authors' interpretations were a stretch. It is first hypothesized on page 7 "...that tumors may arise from a low ploidy state and that the ploidy increase caused by PIDDosome deficiency at the time of weaning may be protective." The data demonstrate the percentage of binucleate hepatocytes is reduced in tumor (T) tissue compared to non-tumor (NT) tissue for each genotype, and this is confirmed by enrichment of cells with lower ploidy nuclei in tumor tissue. The section concludes, "This supports the idea that HCC initiates from cells with low ploidy levels which may have a higher risk of loss of heterozygosity (LOH), and thus, are more likely to transform." WT and KO mice are all injected with DEN at P15, an age before physiological polyploidization begins and >95% of hepatocytes are diploid; therefore, it is true that diploids are susceptible to DEN-induced damage (supported by Zhang, Dev Cell 2018; Wilkinson, Hepatology 2019; and others). The authors show in Fig. 4B that aneuploidy is equivalent between diploids from WT and Casp2-KO mice (and presumably the other KOs), which argues against baseline LOH differences between WT and the KOs, especially at P15. Even if the KOs had higher baseline LOH, how would increased polyploidization by the KOs after weaning protect from tumorigenesis? Polyploidization would occur after DEN-induced damage/mutation, leading to amplification of any pre-existing LOH and DEN-induced mutation. I believe the interpretation would be different if, for example, WT and KO mice were injected with DEN (after polyploidization occurred). In this situation, increased ploidy in the KOs would be expected to guard against tumorigenesis (as seen in Zhang, Dev Cell 2018; Wilkinson, Hepatology 2019. HOWEVER, after reading the Discussion, particularly the top paragraph on page 12, the authors provided a thoughtful and reasonable explanation for my concerns. I am satisfied with their discussion.

Referee #2:

Sladky et al. have investigated the role of supernumerary centrosome-PIDDosome-p53 axis in HCC development. By applying DEN-induced, HCC model to WT and genetically engineered mice in where formation of Casp2-PIDDosome is impaired, author suggested that hepatocytes with low ploidy is prone to transform to cancer stem cells (CSC) and thereby increasing tumorigenesis. As an underlying mechanism, author suggested that ability of Casp2-PIDDosome-p53 to restrict ploidy, shown in author's previous work (Sladky et al., Dev. Cell, 2020), renders DEN-metabolized-, DNA damaged- pericentral hepatocytes to be diploid and stimulates transformation to CSC. Additionally, authors comprehensively analyzed TCGA data from liver cancer patients and found that Casp2 and PIDD1 is highly abundant in HCC specimens and expressions of Casp2 and PIDD1 are

proportionally increased with Ki67 and co-related with enrichment of cell cycle regulators, including E2F targets, further supporting author's previous and current observations in the clinical setting.

Here, I would like to suggest some additional experiments.

Comment 1. At the 10 months after DEN injection, authors found that tumor incidences and tumor sizes are significantly reduced in the liver of Casp2-, PIDD1-, and RAIDD-knock-out (KO) mice, compared to WT (Fig 1B, C). Authors tested Ki67 in both non-tumor (NT) and tumor (T) area to assess whether hepatocyte proliferation accounts for increased tumorigenesis in the WT liver and found that numbers of Ki67-positive cells in T area are similar across the all genotypes, including WT livers (Fig 2B). Therefore, author suggested that tumor incidence is likely increased in WT, compared to Casp2, PIDD1, or RAIDD KO mice. However, authors have not presented stem cell markers in this study.

Comment 2, Authors have used whole body Casp2, PIDD1 or RAIDD KO mice in this study. Given that tumorigenesis can also be affected by Kupffer cells and plasma TG and Chol are important triggers of Kupffer cell activation. Authors are better to analyze infiltrated macrophages in NT and T area to show that tumorigenesis is caused by intrinsic effects of hepatocytes, where PIDDosome is primarily activated.

Comment 3. To test if low ploidy promotes tumorigenesis, authors analyzed ploidy in NT and T area from 10.5 month old, tumor bearing mice and found that binucleated cell numbers are significantly lower in T, compared NT area, in line with author's point. However, numbers of binucleated cells in T area of Casp2 KO and PIDD1 KO seems to be lower than those of WT tumor (Fig 3 A, C). Additionally, as it is pointed out, RAIDD KO mice does not show difference in numbers of octoploid nuclei even though RAIDD KO mice developed tumors in response to DEN (Fig 1B, D). This may be caused from that analyses have been performed in the mice that were fully developed tumors and presented similar levels of liver sickness, ALT and AST (Suppl. Fig1B).

Comment 3-1. DEN is metabolized by Cyp2E1 and induces DNA damage in peri-central hepatocyte. As a result of this, DEN promotes pericentral p53 activation (Dhar et al., Cancer cell, 2018). PIDD1 is a transcriptional target of p53 in response to DNA damage and engages to divergent signaling pathways: NF- κ B and/or PIDDosome activation. Author mentioned that, at the time of DEN injection, at day 15, most hepatocytes are mononuclei, indicating the absence of supernumerary centrosome (SC)-dependent PIDDosome activation. However, SC may arise during the process of DEN-induced cell death and subsequently following proliferations. Dhar et al. showed that DEN injection also robustly increased pericentral p-AKT activation, which is a major player of polyploidy in hepatocytes (Celton-Morizur et al. Cell Cycle, 2010). In addition, Authors are better to show that Cyp2E1 are equal across all strains used in this study.

Comment 3-2. Author's point is that SC-PIDDosome-p53 restricts ploidy in hepatocytes which harbor DNA damage due to DEN metabolites, resulting in promoting tumor initiation. In such a case, SC-negative, DNA damaged hepatocytes can be avoid of surveillance from PIDDosome-p53 and thereby becoming tumorigenic, even if hepatocytes lacks of PIDDosome. Indeed, this case is represented as tumors found in PIDDosome KO mice. Dhar et al. showed that DEN injection into adult mice also is able to test DEN induced cell death and compensatory proliferation and recently published, author's paper (Sladky et al, Dev Cell, 2020) showed that tetraploid nuclei are appeared at day 25, after weaning at day 20, across all genotypes and extent of ploidy is gradually increased in PIDDosome KO mice upto 15 months. Therefore, it is constructive to re-evaluate effects of SC-PIDDosome-p53 in DEN-injected adult WT, Casp2, PIDD1, and RAIDD KO mice by investigating CSC

markers, NF-kB signaling, p53 and p21.

Comment 4. p53 is suggested to suppress a mevalonate pathway, SREBP2-dependent cholesterol synthesis, in tumor cells (Moon et al, Cell, 2019). Author showed that several pathways are negatively or positively co-related with Casp2, RAIDD, and PIDD1-abundant HCC (Fig 6). Adding mevalonate pathway, if possible, to TCGA analyses would be more informative.

Comments on minor points.

1. Figure legends-Fig 2: Casp2 and Raidd are for the mouse, CASP2 and RAIDD are for the human.
2. Figure legends-Fig 3: Author states non-tumorous (NT) and tumor (T) in main body but it is stated as non-tumor (NT) and tumor (T) in figure legend 3.
3. Suppl Fig1E is better to be rearranged to Suppl Fig 2.
4. Fig 3A, D: Please add statistical results between tumors of different genotypes and add N.S if it is non-significant.

Referee #3:

In this paper, Sladsky et al investigate the role of the PIDDosome (PIDD1 and RAIDD regulating Caspase 2). They use carcinogenesis to induce liver tumours in wild-type mice and mice with individual PIDDosome components knocked out. To their surprise, knockout mice develop fewer (but heavier) tumours than wt mice. They found wt tumours had elevated levels of Caspase 2 and PIDD1 but not RAIDD. Ki67 levels were higher in tumours than normal tissues, but this did not consistently vary between wt or knockout tissues, thus it is unclear why wt tumours are smaller but more frequent than knockout tumours. All tumours trended to lower ploidy (as measured by fewer binucleate cells, and fewer 8c genome content cells) than normal tissue. They suggest that Caspase 2 normal tissue cells were more likely to be polyploid than the wt normal tissue cells, and that in line with previous findings that mouse HCC develop from diploid cells, higher polyploidy may provide a barrier to transformation in the liver. They found no link between aneuploidy and caspase 2 activity in polyploid cells suggesting the loss of Caspase 2 was not promoting higher aneuploidy. They then turn to human liver cancer samples and again find increased Caspase 2 and PIDD1 levels in tumours but not RAIDD. Caspase 2 and PIDD levels seem to increase as disease progresses but RAIDD decreases. This relationship is even higher in p53- patients. They found some correlation between Caspase 2 and PIDD expression with cell cycle genes, but not for RAIDD. Overall their conclusion is that their initial conclusion that PIDDosome functions to provide protection against cancer due to increased p53 activity seems to be wrong, and that tumours seem to be less polyploid than normal tissue. This in turn contrasts with a prevailing dogma concerning the potential role of polyploidisation in carcinogenesis.

The role of PIDDosome in tumorigenesis was certainly worthy of investigation but I am not convinced by author conclusions. Overall there is a lack of clarity in the rationale and discussion of results that makes determining their conclusions difficult. No mechanism or clear model to explain the data as a whole are presented. As a consequence I recommend a re-assembly of this data and the conclusions before publication would be suitable.

Major points:

1. Furthermore differences between the outcomes of depleting each of the three PIDDosome components makes interpreting their data difficult in the context of a role for the PIDDosome as a

whole. For example, Caspase 2 has functions independent of PIDDosome so why use it as a surrogate for PIDDosome loss?

2. Authors say that "Extra centrosomes are presumably lacking" when DEN was added to induce DNA damage and promote tumorigenesis, but no centrosome quantification is done at any stage in any of the genotypes in the mice.

3. Figure 3 - tumours seem to have fewer polyploid cells than normal tissues based on binucleate counting, and a weak correlation of reduced % of 8c cells. But there isn't a major difference between the genotypes, so I am not convinced this is linked to why wt tumours are smaller but more frequent.

4. The PIDDosome supposedly regulates p53 through MDM2. No data at all on MDM2 or p53 status in any of the mice tumours by genome sequencing or by RNA expression or western blot?

5. Figure 5B - they say that there is a broad trend in increasing Caspase2 and PIDD expression and progression of disease in humans, and this is more pronounced in the p53 mutants. In 5C they suggest higher expression correlates with worse prognosis - if they further stratify according to p53 status, does p53 loss link to worse prognosis?

6. Caspase 2 has the most frequent effect, with PIDD having weaker effects and RAIDD often not involved, and it seems to be unclear what the role of p53 is in this circumstance. Are they sure this is really Caspase 2 role in PIDDosome regulation of p53? Could it not be a Caspase 2 function that is independent of PIDDosome? Might be useful to try the mouse knockout (at least the caspase 2) in p53 knockout background? Why do they think there is such a discrepancy with RAIDD? It often has no effect or the opposite effect of the PIDD1 and Caspase 2 data? Or perhaps double knockout of MDM2? They say that RAIDD is involved in metabolism.

7. If they would like to show a role for PIDDosome-loss induced elevated ploidy as protective against tumour formation, could they experimentally induce polyploidy before DEN by alternative method, and determine if that protects against tumorigenesis independently of PIDDosome?

Minor points

1. The discussion is very long and in places repetitive.

Referee #1:

Overall, this is a potentially very interesting paper. It is well written, experiments are high quality and use some novel approaches, statistics are appropriate. The topic is timely and relevant. There are number of issues that dampen my enthusiasm. Most significantly, it is unclear how PIDDosome affects tumorigenesis as the links between PIDDosome, ploidy and tumors are unclear. Specific concerns - major and minor - are outlined below and should be addressed.

MAJOR CONCERNS:

1. The authors compared tumorigenesis using the established DEN model and found that germline loss of Casp2, Pidd1 or Raidd resulted in formation of fewer tumors and smaller tumors, although LW/BW remained equivalent. Overall, analysis of the mice and tumors after 10 months is well done. There is a lot of literature describing the types of tumors generated in the DEN model. Does loss of Caspase 2 or PIDDosome components fundamentally alter tumorigenesis - in other words, does the spectrum of tumors in each of the KOs differ from WT?

Response: We appreciate this question. In fact, all mouse tumors have been evaluated by a certified pathologist who identified them as multifocal or focal HCC. One adenoma was found in RAIDD-mutant mice. The incidence of pre-neoplastic lesions and associated adenomas was highest in the wt. This data is now presented in EV Fig. 1B-C.

2. Using the TCGA database, the authors show in Fig. 5 that CASP2 and PIDD1 expression correlates with HCC severity, and that patients with low CASP2 expression have modest, yet statistically significant, improvement in recurrence-free survival compared to patients with high CASP2 expression. Moreover, analysis showed that CASP2 and PIDD1 expression correlate with proliferation-associated gene signatures. These observations are important as they essentially replicate the findings in mice and strongly support a role for CASP2 and PIDD1 in HCC formation. My major concern with this section is the author's attempt to link PIDDosome expression with ploidy alterations to tumorigenesis. The concluding sentence on page 19 is especially confusing,

"Thus, the impact of PIDDosome on hepatocarcinogenesis in mice and the recurrence-free survival in HCC patients is most likely not related to direct effects of the on tumor cell proliferation but rather linked to the key role of the PIDDosome controlling ploidy." The logic needs to be clarified or the conclusion needs to be revised.

Response: We have rephrased our concluding sentence to "Thus, the impact of the PIDDosome on DEN-driven tumor formation in mice and the recurrence-free survival in HCC patients is most likely linked to its key-role in controlling cellular ploidy and not to tumor cell proliferation per se" and hope that this will make our message clearer.

3. Based on data in Fig. 7, the authors suggest the exciting idea that tumor cell density can be used as a marker for recurrence-free survival in HCC patients. Basically, low cell density (and, thus, high ploidy ... see minor comment #3 below) have longer recurrence-free survival. This is supported by mouse studies (e.g., Zhang, Dev Cell 2018; Wilkinson, Hepatology 2019; others). This finding contrasts with Bou-Nader, et al (Gut 2019) where patient HCC are enriched with mononuclear polyploid hepatocytes, especially those with p53 mutations. The authors address this issue in the Discussion, citing the increased patient "n" and the lack of p53 mutational status in the current study. Despite this reasonable argument, I'm left wondering which set of observations/conclusions is correct. Any extra data the authors can provide would tremendously valuable.

Response: We agree with this referee's statement that it is hard to unify the observations made of Bou-Nader et al and our findings. As pointed out, we are not able to define the p53 mutational status in our patient cohort. We are only able to provide additional information on patient characteristics related to the number of tumor nodules and vascular invasion and split them according to the median ploidy level. This information is now included in the new version of Table S1.

Interestingly though, when re-analyzing TCGA data and taking the p53 status into account for recurrence-free survival, as shown in Figure 5B-C and Fig. EV5D, p53 mutant patients with high caspase-2 levels (low ploidy) show much worse prognosis than those with low caspase-2 levels (high ploidy). This finding is in support of our own observations, but, again, arguing against the findings by Bou-Nader. Yet, it is clear that p53 mutant patients do have a generally poor prognosis and a subset of those with high ploidy may have evolved more aggressive tumors. In the end, we cannot provide a satisfying answer, but add to the discussion that additional analysis of PIDDosome component expression levels in patient cohorts with known p53 and ploidy status is required to reconcile these opposing results.

MINOR CONCERNS:

1. The method to measure CNV in "mini bulks" from tumor tissue is a clever approach. I can see two limitations, which should be discussed. First, only clonal aneuploidy should be detected with this approach. Random CNV by any of the 30 cells may be below the limit of detection and, therefore, missed. Second, since the nuclei are isolated from tissue, any nuclei from non-parenchymal cells, inflammatory cells, etc. would contaminate the prep, likely skewing the pool toward euploidy. This should not be an issue with the 4c cells but could affect the 2c cells. Despite these limitations, the CNV patterns between WT and Casp2-KO are similar, and I think the data are sound.

Response: We agree with this referee's conclusion.

2. In Fig. 5A, the pattern of proCASP2 in NT and T tissue is easy to see, but it is harder to appreciate the pattern with PIDD1 expression, which is more heterogeneous. It would be helpful to quantify the data and display as a graph.

Response: We have performed densitometric quantification of our western blot data now presented in Figure EV4A which clearly supports the overall conclusion, i.e. Casp2 and PIDD1 expression are increased in tumor vs. normal tissue.

3. The approach in Fig. 7 to use cell density as readout for ploidy is potentially very useful. Unfortunately, this approach is never verified. The authors should compare the cell density approach with the gold standard method for determining ploidy - FACS analysis. I would suggest comparing ploidy in WT and one or more of the KO mice by both methods. Another useful analysis would be to determine the ploidy index in mouse NT and T tissues and compare with FACS data in Fig. 3.

Figure 1 for internal review only: Linear regression and Pearson correlation of the tumor density with the cumulative diameter or the number of surface tumors in DEN-induced HCC bearing wt mice.

Unfortunately, due to the mouse cohort size, the number of data points is not sufficient to provide a significant result. Yet, we think that the trend seen in Figure 1 here is supporting our conclusions.

4. After reading the Results surrounding Figure 3, I initially felt the authors' interpretations were a stretch. It is first hypothesized on page 7 "...that tumors may arise from a low ploidy state and that the ploidy increase caused by PIDDosome deficiency at the time of weaning may be protective." The data demonstrate the percentage of binucleate hepatocytes is reduced in tumor (T) tissue compared to non-tumor (NT) tissue for each genotype, and this is confirmed by enrichment of cells with lower ploidy nuclei in tumor tissue. The section concludes, "This supports the idea that HCC initiates from cells with low ploidy levels which may have a higher risk of loss of heterozygosity (LOH), and thus, are more likely to transform." WT and KO mice are all injected with DEN at P15, an age before physiological polyploidization begins and >95% of hepatocytes are diploid; therefore, it is true that diploids are susceptible to DEN-induced damage (supported by Zhang, Dev Cell 2018; Wilkinson, Hepatology 2019; and others).

The authors show in Fig. 4B that aneuploidy is equivalent between diploids from WT and Casp2-KO mice (and presumably the other KOs), which argues against baseline LOH differences between WT and the KOs, especially at P15. Even if the KOs had higher baseline LOH, how would increased polyploidization by the KOs after weaning protect from tumorigenesis? Polyploidization would occur after DEN-induced damage/mutation, leading to amplification of any pre-existing LOH and DEN-induced mutation.

I believe the interpretation would be different if, for example, WT and KO mice were injected with DEN (after polyploidization occurred). In this situation, increased ploidy in the KOs would be expected to guard against tumorigenesis (as seen in Zhang, Dev Cell 2018; Wilkinson, Hepatology 2019. HOWEVER, after reading the Discussion, particularly the top paragraph on page 12, the authors provided a thoughtful and reasonable explanation for my concerns. I am satisfied with their discussion.

Response: Thank you very much for your in-depth analysis and thorough review

Referee #2:

Sladky et al. have investigated the role of supernumerary centrosome-PIDDosome-p53 axis in HCC development. By applying DEN-induced, HCC model to WT and genetically

engineered mice in where formation of Casp2-PIDDosome is impaired, author suggested that hepatocytes with low ploidy is prone to transform to cancer stem cells (CSC) and thereby increasing tumorigenesis. As an underlying mechanism, author suggested that ability of Casp2-PIDDosome-p53 to restrict ploidy, shown in author's previous work (Sladky et al., Dev. Cell, 2020), renders DEN-metabolized-, DNA damaged- pericentral hepatocytes to be diploid and stimulates transformation to CSC. Additionally, authors comprehensively analyzed TCGA data from liver cancer patients and found that Casp2 and PIDD1 is highly abundant in HCC specimens and expressions of Casp2 and PIDD1 are proportionally increased with Ki67 and co-related with enrichment of cell cycle regulators, including E2F targets, further supporting author's previous and current observations in the clinical setting.

Comment 1. At the 10 months after DEN injection, authors found that tumor incidences and tumor sizes are significantly reduced in the liver of Casp2-, PIDD1-, and RAIDD-knock-out (KO) mice, compared to WT (Fig 1B, C). Authors tested Ki67 in both non-tumor (NT) and tumor (T) area to assess whether hepatocyte proliferation accounts for increased tumorigenesis in the WT liver and found that numbers of Ki67-positive cells in T area are similar across the all genotypes, including WT livers (Fig 2B). Therefore, author suggested that tumor incidence is likely increased in WT, compared to Casp2, PIDD1, or RAIDD KO mice. However, authors have not presented stem cell markers in this study.

Response: As this referee rightly summarizes, we propose that the tumor incidence is increased in wt over PIDDosome mutant mice because of ploidy differences in the premalignant cell pool and not because of differences in the proliferative capacity of transformed cells expressing or lacking the PIDDosome. We have not addressed the frequency of cancer stem cells, e.g. based on EpCam, CD13, CD90, CD133, surface expression, in our tumor specimens, as all our data suggests that it is the basal ploidy level that limits HCC formation, as recently shown also by others; Zhang et al. 2018. *However, to exclude that the number of CSC is different between wt and PIDDosome deficient tumors, mRNA levels of Thy1 (CD90) and Prom1 (CD133) were compared by RT-qPCR, but failed to reveal major differences. Please note that the significant difference in Thy1 mRNA levels depends on only one Casp2^{-/-} value. We hence conclude that the frequency of CSC is comparable across genotypes.* This data is now shown in Fig. EV2C.

Comment 2, Authors have used whole body Casp2, PIDD1 or RAIDD KO mice in this study. Given that tumorigenesis can also be affected by Kupffer cells and plasma TG and Chol are important triggers of Kupffer cell activation. Authors are better to analyze infiltrated macrophages in NT and T area to show that tumorigenesis is caused by intrinsic effects of hepatocytes, where PIDDosome is primarily activated.

Response: According to our histopathology report for our recent study published in Dev Cell earlier this year and this manuscript, an alteration of Kupffer cell frequency was not evident. Moreover, the frequency of immune cells found infiltrating the tumors did not reveal major differences across genotypes, yielding a comparable histopathological inflammatory score – see new Fig. EV1E-F. Plasma cells were only found infiltrating RAIDD-deficient tumors, yet the significance of this observation remains uncertain. Hence, we conclude that the inflammatory status does not substantially differ across genotypes and hence most likely does not contribute to the noted phenomena reported here.

Comment 3. To test if low ploidy promotes tumorigenesis, authors analyzed ploidy in NT and T area from 10.5 month old, tumor bearing mice and found that binucleated cell numbers are significantly lower in T, compared NT area, in line with author's point. However, numbers of binucleated cells in T area of Casp2 KO and PIDD1 KO seems to be lower than those of WT tumor (Fig 3 A, C). Additionally, as it is pointed out, RAIDD KO mice does not show

difference in numbers of octoploid nuclei even though RAIDD KO mice developed tumors in response to DEN (Fig 1B, D). This may be caused from that analyses have been performed in the mice that were fully developed tumors and presented similar levels of liver sickness, ALT and AST (Suppl. Fig1B).

Response: We appreciate this concern. Albeit the number of binucleated cells appears to be lower in C2 and PIDD1 KO tumors compared to WT, this difference is not statistically different, as indicated now in the corresponding Figure panel. Admittedly, we also have currently no explanation as to why RAIDD-deficient tumors behave differently and will subject this matter to a detailed follow up investigation.

Comment 3-1. DEN is metabolized by Cyp2E1 and induces DNA damage in peri-central hepatocyte. As a result of this, DEN promotes pericentral p53 activation (Dhar et al., Cancer cell, 2018). PIDD1 is a transcriptional target of p53 in response to DNA damage and engages to divergent signaling pathways: NF- κ B and/or PIDDosome activation. Author mentioned that, at the time of DEN injection, at day 15, most hepatocytes are mononuclei, indicating the absence of supernumerary centrosome (SC)-dependent PIDDosome activation. However, SC may arise during the process of DEN-induced cell death and subsequently following proliferations. Dhar et al. showed that DEN injection also robustly increased pericentral p-AKT activation, which is a major player of polyploidy in hepatocytes (Celton-Morizur et al. Cell Cycle, 2010). In addition, Authors are better to show that Cyp2E1 are equal across all strains used in this study.

Response: Indeed, it is important to exclude the possibility that DEN is metabolized at different rates across genotypes, yet, our comparison of Cyp2E1 mRNA expression levels in wt vs. PIDD1-deficient livers at the time of DEN injection (Figure EV2E) raises confidence that this is not the case. Moreover, our analysis of centrosome number before and after DEN injection on day 15 and two days later does not show differences between genotypes – hence we conclude that the PIDDosome is not engaged after DEN injection during DNA damage, but only at the time of weaning (Figure 2C-G).

Comment 3-2. Author's point is that SC-PIDDosome-p53 restricts ploidy in hepatocytes which harbor DNA damage due to DEN metabolites, resulting in promoting tumor initiation. In such a case, SC-negative, DNA damaged hepatocytes can be avoid of surveillance from PIDDosome-p53 and thereby becoming tumorigenic, even if hepatocytes lacks of PIDDosome. Indeed, this case is represented as tumors found in PIDDosome KO mice. Dhar et al. showed that DEN injection into adult mice also is able to test DEN induced cell death and compensatory proliferation and recently published, author's paper (Sladky et al, Dev Cell, 2020) showed that tetraploid nuclei are appeared at day 25, after weaning at day 20, across all genotypes and extent of ploidy is gradually increased in PIDDosome KO mice up to 15 months. Therefore, it is constructive to re-evaluate effects of SC-PIDDosome-p53 in DEN-injected adult WT, Casp2, PIDD1, and RAIDD KO mice by investigating CSC markers, NF- κ B signaling, p53 and p21.

Response: Based on the above (response to comment #1), we are confident that the number of CSC is not different between wt and PIDDosome mutant mice. Moreover, in light of the comments of referee #3, we decided to evaluate the p53 response after DEN injection at the time used to drive HCC, i.e. 15 days, rather than in adult mice. The corresponding data is now shown in Figure 2 and supports the notion that the PIDDosome is not engaged in response to DEN-induced DNA damage but only at the time of weaning, when only residual DNA damage, inflicted by DEN treatment, is still detectable. Moreover, extra centrosomes werenot after DEN injection, excluding the cue for PIDDosome activation being present in DEN-treated hepatocytes.

Comment 4. p53 is suggested to suppress a mevalonate pathway, SREBP2-dependent cholesterol synthesis, in tumor cells (Moon et al, Cell, 2019). Author showed that several pathways are negatively or positively co-related with Casp2, RAIDD, and PIDD1-abundant HCC (Fig 6). Adding mevalonate pathway, if possible, to TCGA analyses would be more informative.

Response: we have performed KEGG pathway analysis, as the Hallmark pathway tool does not have a set of genes corresponding to the mevalonate pathway, to correlate PIDDosome expression levels with enzymes needed for terpenoid backbone synthesis needed for mevalonate synthesis. This analysis did not reveal a significant co-expression of PIDDosome components with genes needed for terpenoid backbone synthesis.

Figure (A) for this reviewer only shows the KEGG pathways with the highest overlap of genes co-expressed with PIDDosome components. Proteins of the mevalonate pathway were not amongst the genes with a correlation of $r > 0.5$ and a p-value < 0.05 . (B) The „KEGG TERPENOID BACKBONE BIOSYNTHESIS“ contains components of the mevalonate pathway. Only one gene shows an $r > 0.5$, however the p-value is not below 0.05. Based on these bioinformatics analyses, we conclude that the mevalonate pathway is not co-regulated with PIDDosome components in HCC.

Figure 2 for internal review only: KEGG pathway analysis of the genes co-expressed with CASP2, RAIDD, and PIDD1.

Comments on minor points.

1. Figure legends-Fig 2: Casp2 and Raidd are for the mouse, CASP2 and RAIDD are for the human.

According to the HUGO nomenclature, it is capital letter for both, mouse and human protein.

2. Figure legends-Fig 3: Author states non-tumorous (NT) and tumor (T) in main body but it is stated as non-tumor (NT) and tumor (T) in figure legend 3.

Corrected

3. Suppl Fig1E is better to be rearranged to Suppl Fig 2.

The Ki67 data are now found in Fig. EV2B.

4. Fig 3A, D: Please add statistical results between tumors of different genotypes and add N.S if it is non-significant.

All necessary comparisons have been tested and non-significant results were indicated where deemed informative. This information has been added to the figure, otherwise, it is stated in the figure legend. Additionally, this is now stated in the methods section "data analysis".

Referee #3:

The role of PIDDosome in tumorigenesis was certainly worthy of investigation but I am not convinced by author conclusions. Overall there is a lack of clarity in the rationale and discussion of results that makes determining their conclusions difficult. No mechanism or clear model to explain the data as a whole are presented. As a consequence I recommend a re-assembly of this data and the conclusions before publication would be suitable.

Major points:

1. Furthermore, differences between the outcomes of depleting each of the three PIDDosome components makes interpreting their data difficult in the context of a role for the PIDDosome as a whole. For example, Caspase 2 has functions independent of PIDDosome so why use it as a surrogate for PIDDosome loss?

Response: This concern is justified to some degree. However, regarding ploidy control, we know from our previous studies (Fava et. al, 2017 G&D, Sladky et. al, 2020 Dev Cell), that loss of Caspase-2 phenocopies the loss of RAIDD and PIDD1, hence, in this context, loss of Casp2 can be seen as equivalent to loss of the PIDDosome. We have included now a sentence on page 10 of the results part to highlight this fact: "Aside from its function within the PIDDosome for liver ploidy control,....."

2. Authors say that "Extra centrosomes are presumably lacking" when DEN was added to induce DNA damage and promote tumorigenesis, but no centrosome quantification is done at any stage in any of the genotypes in the mice.

Response: We agree, the initial version of our work was falling short in this analysis that has now been performed before and after DEN injection. No differences in centrosome numbers were observed prior the time of weaning. The new piece of data is now included in Figure 2.

3. Figure 3 - tumours seem to have fewer polyploid cells than normal tissues based on binucleate counting, and a weak correlation of reduced % of 8c cells. But there isn't a major difference between the genotypes, so I am not convinced this is linked to why wt tumours are smaller but more frequent.

Response: From this comment, we conclude that the way of presenting our data was suboptimal. The figure panels have now been rearranged to make these differences better visible. Moreover, using one-way Anova for data analysis indicates highly significant differences between wt, Casp2 and Pidd1 mutant tumors. Raidd mutant tumors fall out of the equation for reasons unclear, but this fact is discussed in depth on pages 7 and 10.

4. The PIDDosome supposedly regulates p53 through MDM2. No data at all on MDM2 or p53 status in any of the mice tumours by genome sequencing or by RNA expression or western blot?

Response: To address this justified concern, we now include western blot for p53 on wt non-tumorous and tumor tissue (Fig. 2A). Additional data on the human HCC cell line SNU182, showing MDM2 processing and p53 activation upon cytokinesis failure induced by Aurora B inhibition is provided (Fig. EV4B). Moreover, an in-depth analysis of the p53 response after DEN injection and at the time of weaning – the corresponding data is shown in Figure 2D

and supports our conclusion that PIDD1 loss has no impact on DNA-damage induced p53 activation, but only is engaged after weaning.

5. Figure 5B - they say that there is a broad trend in increasing Caspase2 and PIDD expression and progression of disease in humans, and this is more pronounced in the p53 mutants. In 5C they suggest higher expression correlates with worse prognosis - if they further stratify according to p53 status, does p53 loss link to worse prognosis?

Response: As this referee predicts correctly, stratifying according to p53 status increases the spread in disease-free survival when looking at caspase-2 expression levels. A similar result was obtained for Ki67 expression. This additional information is now provided in Figure EV4D. Loss of RAIDD shows again the opposite trend, while this stratification has no impact on the correlation between PIDD1 mRNA and survival, as it has no impact before stratification. However, we did not include these data, as only caspase-2 expression levels alter survival.

6. Caspase 2 has the most frequent effect, with PIDD having weaker effects and RAIDD often not involved, and it seems to be unclear what the role of p53 is in this circumstance. Are they sure this is really Caspase 2 role in PIDDosome regulation of p53? Could it not be a Caspase 2 function that is independent of PIDDosome? Might be useful to try the mouse knockout (at least the caspase 2) in p53 knockout background? Why do they think there is such a discrepancy with RAIDD? It often has no effect or the opposite effect of the PIDD1 and Caspase 2 data? Or perhaps double knockout of MDM2? They say that RAIDD is involved in metabolism.

Response: We appreciate this concern and often struggle ourselves with the differences seen in RAIDD mutant mice which clearly warrants a more detailed follow up. However, our previous analysis of P53/Casp2 double-mutant hepatocytes (Fava et al, 2017 G&D) showed a clear epistatic relationship between Casp2 and p53, arguing that, at least in ploidy control, caspase-2 is upstream of p53 and not redundant with other signaling pathways, or that caspase-2 may be activated in a PIDDosome independent manner in this context. Please note that p53 mutant mice develop thymic lymphomas earlier in life than DEN causes HCC, precluding their analysis. However, spontaneous lymphomagenesis caused by p53 loss is not affected by additional loss of caspase-2 (Manzl et. al, CDD 2012).

7. If they would like to show a role for PIDDosome-loss induced elevated ploidy as protective against tumour formation, could they experimentally induce polyploidy before DEN by alternative method, and determine if that protects against tumorigenesis independently of PIDDosome?

Response: These experiments have been conducted already by others, cited, in support of our findings, by either shifting the time of weaning (Zhang et al. 2018, Dev Cell) followed by DEN injection or by interfering with cytokinesis by performing Anilin knockdown in hepatocytes (Zhang et al. 2018 Gastroenterology, Lin et. al, 2020 Gastroenterology).

Minor points

1. The discussion is very long and in places repetitive.

We revised and shortened the discussion, hoping to make it an easier read.

Dear Andreas,

Thank you for the submission of your revised manuscript to our editorial offices. We have now received the reports from the two referees that were asked to re-evaluate your study, you will find below. As you will see, the referees now support the publication of your study in EMBO reports. However, referees #1 and #3 have some remaining points and some suggestions to improve the study, we ask you to address in a final revised manuscript.

- I wonder if we could have a more comprehensive title. How about:

PIDDosome-induced p53-dependent ploidy restriction facilitates hepatocarcinogenesis

- Please add scale bars of the same style to all the microscopic images. Do not write on the scale bars. Please provide the size information in the respective legend. Please add scale bars also to Figs. EV1A and EV5B/C.

- For Fig. 2F, could you please separate the images with white lines to indicate the single images the panel is composed of?

- Please note our new reference format. Please format your reference list accordingly:
<http://www.embopress.org/page/journal/14693178/authorguide#referencesformat>

- Could the information about the use of Biorender.com be moved to the acknowledgements?

- It seems Gerald Timelthaler, Kaoru Tsuchia and Matthias Pinter are missing from the author contributions. Please check.

- There are callouts to Fig S5, which should be EV5, I guess. Please check.

- As the Western blots shown are significantly cropped, could you provide the source data for all the blots (main and EV figures). The source data will be published in a separate source data file online along with the accepted manuscript and will be linked to the relevant figure. Please submit the source data (scans of entire blots) together with the revised manuscript. Please include size markers for scans of entire blots, label the scans with figure and panel number and send one PDF file per figure.

- Finally, please find attached a word file of the manuscript text (provided by our publisher) with changes we ask you to include in your final manuscript text, and some queries, we ask you to address. Please provide your final manuscript file with track changes, in order that we can see any modifications done.

In addition, I would need from you:

- a short, two-sentence summary of the manuscript

- two to three bullet points highlighting the key findings of your study

- a schematic summary figure (in jpeg or tiff format with the exact width of 550 pixels and a height of not more than 400 pixels) that can be used as a visual synopsis on our website.

Kind regards,

Achim

Achim Breiling
Editor
EMBO Reports

Referee #1:

The authors satisfactorily addressed most of my comments. However, the following two concerns still need to be addressed.

MINOR CONCERN #1

My initial concern - "The method to measure CNV in "mini bulks" from tumor tissue is a clever approach. I can see two limitations, which should be discussed. First, only clonal aneuploidy should be detected with this approach. Random CNV by any of the 30 cells may be below the limit of detection and, therefore, missed. Second, since the nuclei are isolated from tissue, any nuclei from non-parenchymal cells, inflammatory cells, etc. would contaminate the prep, likely skewing the pool toward euploidy. This should not be an issue with the 4c cells but could affect the 2c cells. Despite these limitations, the CNV patterns between WT and Casp2-KO are similar, and I think the data are sound."

Authors' response - "We agree with this referee's conclusion."

The issue - The caveats of the authors' approach need to be described in the text.

MINOR CONCERN #3

My initial concern - "The approach in Fig. 7 to use cell density as readout for ploidy is potentially very useful. Unfortunately, this approach is never verified. The authors should compare the cell density approach with the gold standard method for determining ploidy - FACS analysis. I would suggest comparing ploidy in WT and one or more of the KO mice by both methods. Another useful analysis would be to determine the ploidy index in mouse NT and T tissues and compare with FACS data in Fig. 3."

Authors' response - "Figure 1 for internal review only: Linear regression and Pearson correlation of the tumor density with the cumulative diameter or the number of surface tumors in DEN-induced HCC bearing wt mice. Unfortunately, due to the mouse cohort size, the number of data points is not sufficient to provide a significant result. Yet, we think that the trend seen in Figure 1 here is supporting our conclusions."

The issue - The authors' response is appreciated; however, it does not address my concern. They need to demonstrate that cell density is a valid readout for ploidy (using one of the methods I described, for example).

Referee #2:

In the past months, Sladky et al. conducted additional analyses for CSC markers, liver immune cells and performed mouse experiments, in which WT and PIDD1 knock-out (KO) mice were injected DEN at p15 days and analyzed Casp2PIDDosome and p53 activation in different time courses. These analyses provided results that further assist the role of Casp2PIDDosome in DEN-induced tumorigenesis and support author's original findings.

Author measured transcript levels of CSC markers, CD90 and CD133, in non-tumor or tumor area obtained from DEN-injected, WT and Casp2KO mice and found that CD133 is not significantly changed in Casp2KO tumors compared to WT tumors and CD90 is increased approximately by 3-fold in Casp2KO tumors (Fig. EV2C). However, as the author showed that this tumor is fully developed (Fig. 1 and EV1), the author investigated activation of Casp2PIDDosome and p53 in DEN-injected, young mice.

To test whether Casp2PIDDosome-p53 axis promotes tumor initiation in low-ploidy affiliated hepatocytes, author injected DEN to WT and PIDD1KO mice and analyzed Casp2PIDDosome components and p53 in different time points. In this experiment, author showed that DEN is metabolized by CYP2E1 in both WT and PIDD1-ablated liver in the similar degree (Fig. EV2E) and induced DNA damage and p53 activation comparably (Fig. 2D, t8h and t48h), indicating that PIDDosome is dispensable for DEN-induced DNA damage and p53 induction. Previously, author and other researchers found that hepatocyte polyploidy is the post-weaning event. Therefore, author analyzed Casp2PIDDosome and p53 activation in post-weaned liver and found that p53 is, in part, regulated by PIDD1. Given that p53 in PBS-injected, post-weaned PIDD1KO liver is even lower than one of WT liver (Fig. 2D 5d), this indicates that Casp2PIDDosome-p53 activation is the molecular event, happened restrictively to ploidy-affiliated hepatocytes and author showed that centriole numbers are not affected by DEN-induced DNA damage. Author also found that macrophage population is not affected by DEN-injected liver and overall grade of inflammation is similar across genotypes (Fig. EV1E-F).

In summary, the author provided additional results that further support the author's original finding which PIDDosome restricts ploidy in hepatocyte by activation of p53, thereby promoting tumor initiation. Author showed that PIDDosome-induced p53 stabilization takes place in parallel with DEN-induced DNA damage in post-weaned WT hepatocytes. According to tumor-suppressive role of p53, this WT liver is expected to be protected from HCC progression. However, WT livers showed increased HCC progression, compared PIDD1KO livers, supporting the author's original finding that prolonged p53 activation, acquired by PIDDosome activation, assists tumorigenesis. In this manuscript, with comprehensive analyses of mouse models and human HCC specimens, author suggested a new molecular insight into how p53 plays pleiotropic roles in HCC progression and provided a novel prognostic marker.

Referee #3:

The authors have answered my specific comments satisfactorily. However, I am still concerned about certain aspects of the work, although these should all be addressable with text edits.

1. The fact that RAIDD knock out often does not phenocopy Caspase 2 knock out is still concerning. The authors acknowledge this and discuss possible reasons but I ask that they either explain this discrepancy more clearly, or at least add some additional lines to the discussion to say that their model is not completely clear, and explanations may possibly be due to some other version of the conventional Piddosome complex (perhaps, as we suggested in the original review, more reliant on Caspase 2 and not involving RAIDD).

2. I asked for patient data to be stratified according to p53 status which the authors duly provided (EV4d). However this new data raises an important question: If their hypothesis that Piddosome role in tumorigenesis is via p53, then in p53 mutant patients the presence or absence of piddosome expression might be predicted to be less relevant. In fact, the authors present the opposite trend in extended figure 4d, unless we have misunderstood? Could they clarify how they interpret this data in the context of their hypothesis?

P-T-P

Referee #1

The authors satisfactorily addressed most of my comments. However, the following two concerns still need to be addressed.

MINOR CONCERN #1

My initial concern - "The method to measure CNV in "mini bulks" from tumor tissue is a clever approach. I can see two limitations, which should be discussed. First, only clonal aneuploidy should be detected with this approach. Random CNV by any of the 30 cells may be below the limit of detection and, therefore, missed. Second, since the nuclei are isolated from tissue, any nuclei from non-parenchymal cells, inflammatory cells, etc. would contaminate the prep, likely skewing the pool toward euploidy. This should not be an issue with the 4c cells but could affect the 2c cells. Despite these limitations, the CNV patterns between WT and Casp2-KO are similar, and I think the data are sound."

Reply: We have now addressed this issue in our discussion by adding the following:

Admittedly, our mini-bulk sequencing approach may be geared to detect preferentially clonal aneuploidies and non-hepatocyte nuclei may skew our results in the 2C fraction analyzed. Yet, the number of infiltrating inflammatory cells in tumor tissue is modest and should not substantially alter these results, as also comparable across genotypes (EV1).

MINOR CONCERN #3

My initial concern - "The approach in Fig. 7 to use cell density as readout for ploidy is potentially very useful. Unfortunately, this approach is never verified. The authors should compare the cell density approach with the gold standard method for determining ploidy - FACS analysis. I would suggest comparing ploidy in WT and one or more of the KO mice by both methods. Another useful analysis would be to determine the ploidy index in mouse NT and T tissues and compare with FACS data in Fig. 3."

Authors' response - "Figure 1 for internal review only: Linear regression and Pearson correlation of the tumor density with the cumulative diameter or the number of surface tumors in DEN-induced HCC bearing wt mice. Unfortunately, due to the mouse cohort size, the number of data points is not sufficient to provide a significant result. Yet, we think that the trend seen in Figure 1 here is supporting our conclusions."

The issue - The authors' response is appreciated; however, it does not address my concern. They need to demonstrate that cell density is a valid readout for ploidy (using one of the methods I described, for example).

Reply: *Unfortunately, we are unable to validate that cell density in human tumor tissue is a clear read out for ploidy, as confirmed by flow cytometry. On the one hand, we do not have access to snap frozen tumor tissue of patients analyzed by morphometry in Figure 7 and analysis of the few tumor specimens by mice where a morphometric analysis has also been performed did not reveal a clear-cut result.*

As a detailed follow-up would require additional tumors to be generated in mice or access to human patient material, which will require a substantial amount of time and effort, as well as ethical permission, we propose to address this limitation in our discussion. Hence, we have added the following statement in our discussion:

However, experimental validation of the robustness of tumor cell density as a measure of cellular ploidy in matched patient samples subjected to morphometric as well as flow cytometric analysis is still pending and will be subject of future investigations.

Referee #3:

The authors have answered my specific comments satisfactorily. However, I am still concerned about certain aspects of the work, although these should all be addressable with text edits.

1. The fact that RAIDD knock out often does not phenocopy Caspase 2 knock out is still concerning. The authors acknowledge this and discuss possible reasons but I ask that they either explain this discrepancy more clearly, or at least add some additional lines to the discussion to say that their model is not completely clear, and explanations may possibly be due to some other version of the conventional Piddosome complex (perhaps, as we suggested in the original review, more reliant on Caspase 2 and not involving RAIDD).

Reply: *To address this concern we have added the following statement to the discussion:*

It remains possible that the differences in phenotype observed in the absence of RAIDD are driven by a gain of function of PIDD1, that may then preferentially form alternative signaling complexes, such as the NEMO-PIDDosome, or the recruitment of caspase-2 into alternative signaling platforms.

2. I asked for patient data to be stratified according to p53 status which the authors duly provided (EV4d).

However, this new data raises an important question: If their hypothesis that Piddosome role in tumorigenesis is via p53, then in p53 mutant patients the presence or absence of piddosome expression might be predicted to be less relevant. In fact, the authors present the opposite trend in extended figure 4d, unless we have misunderstood? Could they clarify how they interpret this data in the context of their hypothesis?

Reply: *we note that there is a remaining misconception. Our data suggest that the role of the PIDDosome in HCC is linked solely to its ability to limit hepatocyte ploidy during development. To do so, p53 is engaged. Yet, once tumors arise, the PIDDosome does not affect tumorigenesis directly, but the higher ploidy protects (as also shown by others). At that stage, CASP2 or PIDD1 mRNA expression are merely linked to the proliferation rate of these tumors. Therefore, expression levels of CASP2 increases with stage, assuming that their proliferative capacity increases and is highest in tumors with mutant p53. We have tried to clarify this by updating our results section, now stating:*

This effect is strongly affected by the mutational status of p53, indicating that these tumors are more aggressive. Of note, the disease-free survival of patients with wt p53 was not affected by the CASP2-expression level (Fig. EV4D), suggesting again that higher mRNA levels may be linked solely to higher tumor proliferation rates and are not predictive for disease outcome per se.

Prof. Andreas Villunger
Medical University of Innsbruck
Division of Dev. Immunology
Biocenter
Innrain 80
Innsbruck A-6020
Austria

Dear Prof. Villunger,

I am very pleased to accept your manuscript for publication in the next available issue of EMBO reports. Thank you for your contribution to our journal.

At the end of this email I include important information about how to proceed. Please ensure that you take the time to read the information and complete and return the necessary forms to allow us to publish your manuscript as quickly as possible.

As part of the EMBO publication's Transparent Editorial Process, EMBO reports publishes online a Review Process File to accompany accepted manuscripts. As you are aware, this File will be published in conjunction with your paper and will include the referee reports, your point-by-point response and all pertinent correspondence relating to the manuscript.

If you do NOT want this File to be published, please inform the editorial office within 2 days, if you have not done so already, otherwise the File will be published by default [contact: emboreports@embo.org]. If you do opt out, the Review Process File link will point to the following statement: "No Review Process File is available with this article, as the authors have chosen not to make the review process public in this case."

Should you be planning a Press Release on your article, please get in contact with emboreports@wiley.com as early as possible, in order to coordinate publication and release dates.

Thank you again for your contribution to EMBO reports and congratulations on a successful publication. Please consider us again in the future for your most exciting work.

Yours sincerely,

Achim Breiling
Editor
EMBO Reports

THINGS TO DO NOW:

You will receive proofs by e-mail approximately 2-3 weeks after all relevant files have been sent to our Production Office; you should return your corrections within 2 days of receiving the proofs.

Please inform us if there is likely to be any difficulty in reaching you at the above address at that time. Failure to meet our deadlines may result in a delay of publication, or publication without your corrections.

All further communications concerning your paper should quote reference number EMBOR-2020-50893V3 and be addressed to emboreports@wiley.com.

Should you be planning a Press Release on your article, please get in contact with emboreports@wiley.com as early as possible, in order to coordinate publication and release dates.

Corresponding Author Name: Andreas Villunger

Manuscript Number: EMBOR-2020-50893-T